# The frequency gradient of human resting-state brain oscillations follows cortical hierarchies

**Keyvan Mahjoory[1]\*, Jan-Mathijs Schoffelen[2], Anne Keitel[3], Joachim Gross[1,4,5]\***

[1]Institute for Biomagnetism and Biosignalanalysis (IBB), University of Muenster, Muenster, Germany; [2]Radboud University Nijmegen, Donders Institute for Brain, Cognition and Behaviour, Nijmegen, Netherlands; [3]Psychology, University of Dundee, Scrymgeour Building, Dundee, United Kingdom; [4]Centre for Cognitive Neuroimaging (CCNi), University of Glasgow, Glasgow, United Kingdom; [5]Otto-Creutzfeldt-Center for Cognitive and Behavioral Neuroscience, University of Muenster, Muenster, Germany

**Abstract** The human cortex is characterized by local morphological features such as cortical thickness, myelin content, and gene expression that change along the posterior-anterior axis. We investigated if some of these structural gradients are associated with a similar gradient in a prominent feature of brain activity - namely the frequency of oscillations. In resting-state MEG recordings from healthy participants (N = 187) using mixed effect models, we found that the dominant peak frequency in a brain area decreases significantly along the posterior-anterior axis following the global hierarchy from early sensory to higher order areas. This spatial gradient of peak frequency was significantly anticorrelated with that of cortical thickness, representing a proxy of the cortical hierarchical level. This result indicates that the dominant frequency changes systematically and globally along the spatial and hierarchical gradients and establishes a new structure-function relationship pertaining to brain oscillations as a core organization that may underlie hierarchical specialization in the brain.

**\*For correspondence:**
kmahjoory@gmail.com (KM);
joachim.gross@uni-muenster.de
(JG)

**Competing interests:** The authors declare that no competing interests exist.

## Introduction

It is well established that the brain's cortical areas differ in their cyto- and myeloarchitectonic structure, local and long-range anatomical connectivity, activity and, by consequence, their function (*Glasser et al., 2016*; *Huntenburg et al., 2017*). Interestingly, many structural features that distinguish individual brain areas change gradually in an orderly manner across the cortex, leading to spatial feature gradients. The most prominent and best established gradients are evident along the posterior-anterior axis (*Eickhoff et al., 2018*; *Felleman and Van Essen, 1991*; *Huntenburg et al., 2018*). For instance, neuron density decreases and neuronal connectivity increases from posterior to anterior brain areas. These differences have been attributed to differences in neurogenesis for posterior compared to anterior brain areas (*Hill et al., 2010*; *Huntenburg et al., 2018*). A similar posterior-anterior gradient has been observed for myelin content, cortical thickness, and gene expression (*Burt et al., 2018*). Next to the posterior-anterior gradient, other global spatial organization principles have been proposed to explain the variation of microstructural features across the cortex. For instance, Huntenburg et al. suggest a sensorimotor to transmodal gradient as an important intrinsic organizing dimension of human cortex (*Huntenburg et al., 2018*) reflecting gradual changes in structural features from functionally unimodal (dedicated sensory or motor) areas to higher order, transmodal areas.

In addition to structural gradients as an organizing principle reflecting global cortical organization, it is well acknowledged that cortical areas are structurally connected into larger networks, which often display a hierarchical organization. Cortical hierarchies are typically established based on the degree of microstructural differentiation of the connected areas, and on the classification of the anatomical connections as feedforward or feedback using histological tract-tracing. Early sensory areas with predominantly feedforward outgoing connections are placed at the bottom of the hierarchy and higher order association areas with mostly feedback outgoing connections are placed at the top of the hierarchy (*Felleman and Van Essen, 1991*; *Markov et al., 2014*). A noninvasive, but indirect index of these hierarchies is cortical thickness, a macroscopic feature of the cortex, which can be estimated from MRI scans. It has been shown that cortical thickness mirrors global hierarchical organization of the cortex as well as local hierarchies in visual, auditory and somatosensory areas (*Jasmin et al., 2019*; *Wagstyl et al., 2015*), and, therefore, could be used as a basis for understanding hierarchy-gradient relationships in the cortex.

The presence of these anatomical gradients raises the question to what extent they are reflected in features of brain activity and brain function. Indeed, it has been shown that cortical areas follow a hierarchical ordering in their timescales of intrinsic fluctuations as for example measured in the auto-correlation of spiking activity (*Murray et al., 2014*). Sensory areas show faster fluctuations while frontal areas show slower fluctuations. Shorter timescales in sensory areas enables them to reflect dynamic changes in the environment, whereas the longer timescales in prefrontal areas allows for integration of information. Particularly, this gradient of 'temporal receptive windows' has been demonstrated in visual (*Himberger et al., 2018*) and auditory processing (*Jasmin et al., 2019*) and could be related to the frequency of spontaneous brain oscillations. Oscillations are a prominent feature of brain activity, and have been suggested to play a central role in coordinating neuronal activity (*Fries, 2005*; *Wang, 2010*). Similar to many anatomical features described above, the spectral activity patterns have been shown to be characteristic for each brain area (*Keitel and Gross, 2016*). This is consistent with the view that the individual anatomical structure of a brain area shapes its rhythmic neuronal activity, which led us to hypothesize the existence of a posterior-anterior gradient in the frequency of spontaneous brain rhythms.

Spontaneous rhythms have been studied in the past but typically by focusing on the power in specific frequency bands (*Hillebrand et al., 2016*; *Keitel and Gross, 2016*; *Mellem et al., 2017*). Overall, these MEG studies revealed strongest cortical generators for the dominant alpha rhythm (7–13 Hz) in occipito-parietal brain areas. The beta band (15–30 Hz) shows strongest activity in sensorimotor areas while delta (1–3 Hz) and theta (3–7 Hz) bands are associated with activity in wide-spread areas including frontal cortex.

Here, we adopt a different approach that is based on sophisticated identification of spectral peaks in the power spectra of source-localized resting-state MEG data and included modelling of the 1/f spectral background (*Haller et al., 2018*). This approach offers two distinct advantages. First, focusing on spectral peaks ensures that results are indeed based on brain oscillations. This is not necessarily the case when using the power in a pre-defined frequency band or using band-pass filtered data. Second, by explicitly modeling the 1/f spectral background across the entire cortex we can dissociate contributions due to aperiodic neuronal background activity from those originating from oscillatory activity.

We used this approach to specifically test the hypothesis of a posterior-anterior gradient in the frequency of spontaneous brain rhythms. We identified the frequencies of the dominant brain rhythm across the cortex in source-localized resting-state MEG data of 187 individuals.

As we describe below, we found spatial gradients of the dominant peak frequency across the cortex following the cortical hierarchy.

## Results

### Spatial gradients of the dominant peak frequency of oscillations

We analyzed publicly available resting-state MEG data from 187 participants (*Schoffelen et al., 2019*), reconstructing cortical activity time courses for 7548 dipolar sources located in the cortical surface. We parceled the cortex to 384 regions-of-interest (ROIs) using the cortical parcellation

(introduced in *Schoffelen et al., 2017*) constructed from the Conte69 atlas (*Van Essen et al., 2012*), which divides the cortical surface according to the division introduced by *Brodmann, 1909*.

For each ROI, we concatenated the source time courses of the locations belonging to that ROI and reduced the dimensionality using a singular value decomposition. The obtained components were segmented to 2 s epochs. Power spectra were computed for each epoch, pooled across components, and eventually their 10% trimmed mean was computed to obtain a single spectrum per ROI and individual.

Spectral peaks were identified from ROI spectrum after fitting and subtracting the arrhythmic 1/f component (see *Figure 1A* and Materials and method section). Subsequently, we identified for each participant and ROI the spectral peak with strongest amplitude in the original power spectrum (dominant peak frequency, or for simplicity, merely called peak frequency (PF)). We used PF to test our hypothesis of a posterior-anterior frequency gradient.

*Figure 1B*, top panel, shows the distribution of PF as a function of the ROI's location along the y-axis of the coordinate system (posterior to anterior). Each point represents the trimmed mean across participants of the PF for one ROI. A clear gradual decrease of PF from posterior to anterior is evident and supported by a significant correlation (skipped Pearson correlation, Robust Correlation Toolbox) (*Pernet et al., 2012*) between the trimmed mean PF and the ROI's y-coordinate ($r = -0.84$, $p \ll 0.001$). This frequency gradient is also evident in the cortical maps that show the trimmed mean of the PF across participants for the 384 ROIs (*Figure 1B*, bottom panel). At the individual level, we computed the correlation between the Y-coordinates and PF values and found a significant posterior-to-anterior gradient of PF for 84% of participants ($p < 0.05$). *Figure 1B*, top-right panel, illustrates the distribution of the obtained within-individual correlation values across all individuals and shows consistency but also the variability of this gradient (t-value = $-15.52$, $p < 0.001$) across individuals.

To account for this inter-individual variability and also assess global, consistent, and systematic changes of PF across the cortical surface along the three-dimensional space and along the known established cortical hierarchies, we applied a comprehensive statistical model using mixed-effect modeling. Our use of linear-mixed-effect models (LMEM) ensured that the individual differences are properly accounted for and that significant gradients are consistently present in individual participants. For example, participants have different alpha peak frequency in occipital brain areas and the slope of frequency gradient is different. These individual differences are specifically modelled by LMEM as random effects. Importantly, LMEM applies two-level statistics, and therefore, will only show a significant gradient if it is significant across cortical areas at the individual level, and consistent across participants at the group level. We used PF as the response variable, and the coordinates of the ROI centroids (X: left to right, Y: posterior to anterior, and Z: inferior to superior) plus their two-way interactions set as fixed effects. We modelled the individual slope and offset as random effects to account for variability between participants.

The fixed effect parameters capture mean-variation in the PF that is shared by all individuals (see Materials and methods section), while the participant-unique variance of the PF is addressed by random effects. Thus, our model provides a robust and comprehensive characterization of spatial changes of PF across the cortex and addresses the confound of inter-individual differences. *Figure 1C* displays a table of T-values for fixed-effect parameters of LMEM and the modelled PF on the cortex. LMEM yielded highly significant scores for Y (t = $-15.6$, $p \ll 0.001$), Z (t = $-10.4$, $p \ll 0.001$), and Y:Z (t = $-32$, $p \ll 0.001$) directions. Together, these results support the conclusion that the peak frequency of brain oscillations decreases systematically in posterior-anterior direction. Furthermore, LMEM identified a second global axis where PF decreases along the Z axis (inferior-superior direction).

On the basis of the observed frequency gradient, the question may arise, whether the spatial pattern of frequency across the cortex is the result of spatial leakage originating from an occipital alpha and frontal theta source. If this is the case, we would not expect to see significant frequency change in areas close to primary visual area (V1). To address this question, we computed the geodesic distance between the reference ROI, V1, and all areas located 2–3 cm away from V1, and applied linear mixed effect modelling of PF as a function of the distance values. We found a highly significant negative dependence between PF and distance (t = $-21.1$, $p \ll 0.001$).

To further account for the potential confounding effect of spatial leakage, we performed a new comprehensive analysis, where we computed the geodesic distance between centroid of all ROIs

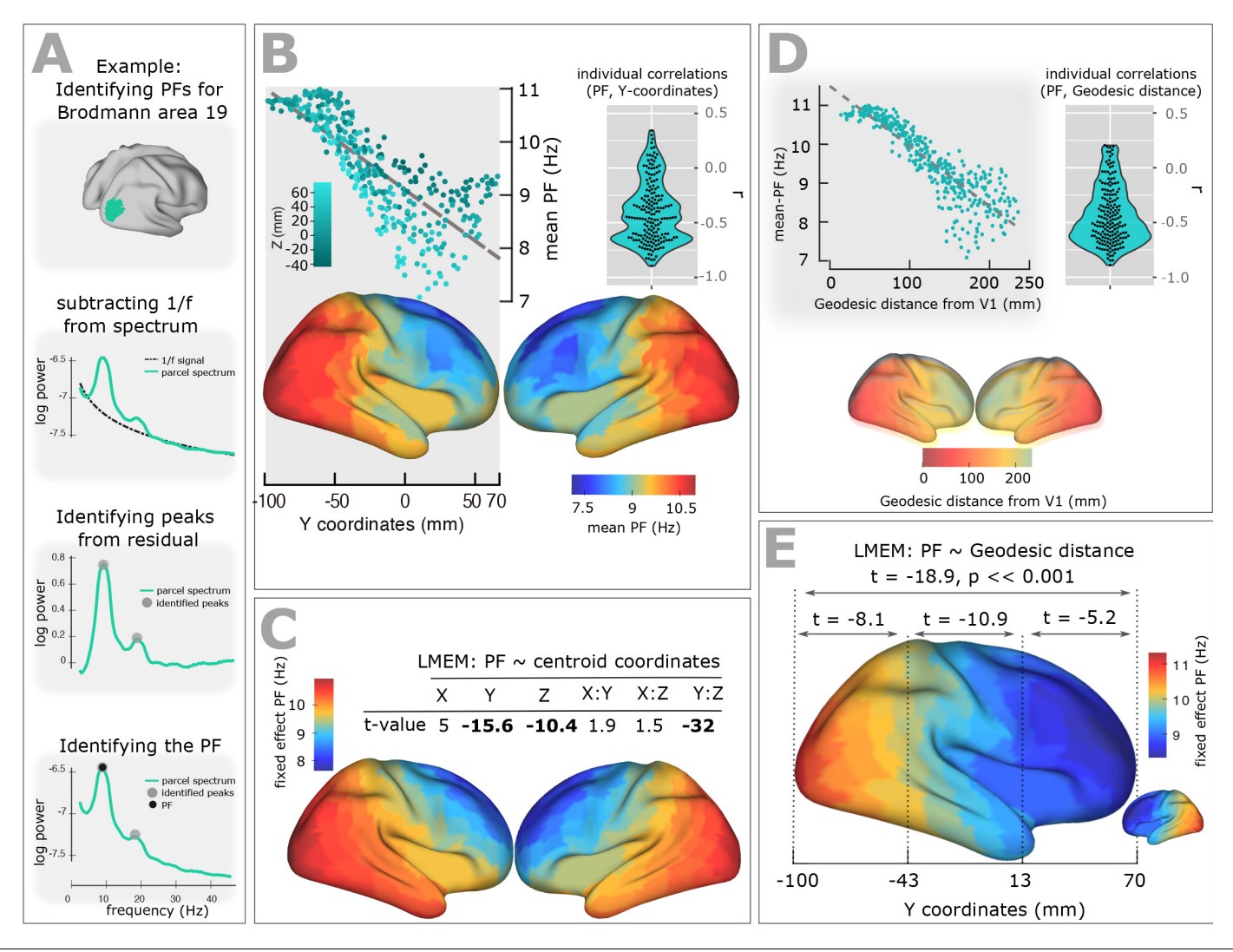

**Figure 1.** Spatial gradient of peak frequency (PF) across the human cortex follows the posterior-anterior hierarchy. (A) Estimating the power spectrum for each cortical region, and identifying peak frequencies after fitting and subtracting the arrhythmic 1/f component. (B) Top-left panel shows the distribution of PF as a function of the ROI's location along the y-axis of the coordinate system (posterior to anterior). Each point represents the trimmed mean across participants of the PF for one ROI (r = −0.84, p<<0.001). Points are colored according to their Z coordinates. Bottom panel: distribution of trimmed mean PFs across 384 cortical ROIs. Top-right panel: Individual level correlation values computed between PF and y-coordinates across ROIs (significant for 84% of participants, p<0.05) and their consistency across all individuals (t-value = −15.52, p<0.001). (C) Top panel: t-values obtained from linear mixed effect modeling of PF as a function of the coordinates of the ROI centroids. Bottom panel: cortical map of the corresponding fixed effect parameters (see *Equation 2* for details). (D) Top-left panel: correlation between trimmed mean PF and geodesic distance (r = −0.89, p<0.001). Top-right panel: individual correlation values (significant for 88% of participants, p<0.05) and their consistency across all individuals (t-value = −17.32, p<0.001). Bottom panel illustrates geodesic distance values mapped on the cortical surface. Geodesic distance was computed between centroid of all ROIs and the centroid of V1 and used this as a new axis to explain the posterior-anterior direction in the cortex. (E) LMEM of PF as a function of geodesic distance performed, separately, for all cortical ROIs, posterior-parietal ROIs (Y < −43 mm), central ROIs (−43 < Y < 13), and frontal ROIs (Y > 13). To assess the distribution of PF along the posterior-anterior axis while accounting for the inter-individual variability, LMEM was applied between the PF as a response variable and the geodesic distance values as an independent variable, and identified a highly significant gradient of PF along the specified geodesic distance (t = −18.9, p<<0.001). Furthermore, to test whether the spatial gradient of PF constantly exists in different areas of the cortex, the cortical surface was split to three equal, consecutive and non-overlapping windows (about 4 cm) based on its Y axis. For each window, LMEM was applied between PF and geodesic distance, and found a significant gradient (window 1: t = −8.1, window 2: t = −10.9, window 3: t = −5.2, all p<0.001). Indeed, our analysis demonstrates a significant organization of PF along the posterior-anterior direction for all windows indicating that this axis constitutes a systematic and constant gradient of PF.

The online version of this article includes the following figure supplement(s) for figure 1:

*Figure 1 continued on next page*

*Figure 1 continued*

**Figure supplement 1.** Stability of the PF gradient over time PF gradient along the posterior-anterior direction computed for 1 st and 2nd halves after splitting the time course to two equal segments.

**Figure supplement 2.** Within- and between-participant variability of ROIs' size and their impact on PF gradient.

and the centroid of V1 (*Figure 1D* bottom panel) and used this as a new axis because it well explains the posterior-anterior axis in the cortex. Our analysis showed a significant negative correlation between the trimmed mean PF and the geodesic distance values (r = −0.89, p<<0.001; *Figure 1D* top-left panel). This negative correlation was evident at the individual level for 88% of participants (p<0.05) and was consistent across individuals (t-value = −17.32, p<0.001; *Figure 1D* top-left panel). To account for this inter-individual variability, we applied an LMEM between the PF as a response variable and the geodesic distance values as an independent variable. We found a highly significant gradient of PF along the specified geodesic distance (t = −18.9, p<<0.001; *Figure 1E*, cortical map). Furthermore, to answer the question whether the spatial gradient of PF constantly exists in different areas of the cortex, we split the cortex to three equal, consecutive and non-overlapping windows (about 4 cm) based on the Y axis, applied LMEM for each window modelling PF as a function of geodesic distance, and found a significant gradient (window 1: t = −8.1, window 2: t = −10.9, window 3: t = −5.2, all p<0.001) (*Figure 1E*). Indeed, our analysis demonstrates a significant organization of PF along the posterior-anterior direction for all windows indicating that this axis constitutes a systematic and constant gradient of PF.

## Spatial gradients of spectral properties of the 1/f Signal

Neurophysiological signals typically consist of oscillatory signal components with distinct spectral peaks, embedded in an arrhythmic 1/f signal component. Variation in the properties of this 1/f component may give rise to shifts of spectral peak estimates, and lead to misidentification of peak frequencies (*Haller et al., 2018*). To investigate this issue, we examined the spatial distribution across the cortex of the estimated slope and offset parameters of the arrhythmic component (see Materials and method section), using LME modeling. As illustrated in *Figure 2A and B*, we found significant scores for Y (slope: t = −4.3, p<<0.001; offset: t = 2.8, p<0.01), Y:Z (slope: t = 6.9, p<<0.001; offset: t = 13.2, p<0.001), and X:Y (slope: t = −6.8, p<<0.001; offset: t = −5.8, p<<0.001) directions. These results indicate a significant decrease of the 1/f slope, and an increase of its offset along the posterior-to-anterior direction. The observed similarity between spatial patterns of 1/f parameters and PF, brings up the question to what extent these parameters could contribute to the observed PF gradient. To assess this, we tested to what extent the spatial change of PF is independent of spatial changes of 1/f slope and offset. We thus used LMEM and regressed out the linear contribution of 1/f slope and offset to PF. Then, we again used LMEM to model the residual PF values as a function of spatial coordinates. The results confirmed a significant posterior-anterior gradient of residual PF values (t-values: Y = −8.3, Z = −4.3, Y:Z = −16; all p<<0.001, *Figure 2C and D*). We therefore conclude that the posterior-anterior PF gradient is largely independent of the observed gradients of slope and offset of the 1/f component.

## Frequency gradients follow cortical hierarchies

The visual system's cortical hierarchy largely progresses along the posterior-anterior direction, and starts in early visual areas in occipital cortex and progresses along the dorsal and ventral streams to anterior areas. Since this progression of cortical hierarchical level coincides with the observed gradient in PF, we tested the hypothesis that the PF gradient is more closely related to cortical hierarchical level than to spatial location. We used cortical thickness (CT) as a proxy for the quantification of the hierarchical level of brain areas (*Valk et al., 2020*; *Wagstyl et al., 2015*).

We used Freesurfer to estimate CT as the shortest distance between corresponding vertices on the white matter surface and the pial surface. To obtain a thickness value for each cortical region, the individual thickness scores were averaged across vertices of that region. *Figure 3* shows a significant change of mean CT along the posterior-anterior axis (r = 0.36, p<<0.001, top panel). The bottom panel of *Figure 3* depicts CT values averaged across participants and mapped on the cortex. LMEM of CT as a function of ROI coordinates showed a significant and progressive increase of CT

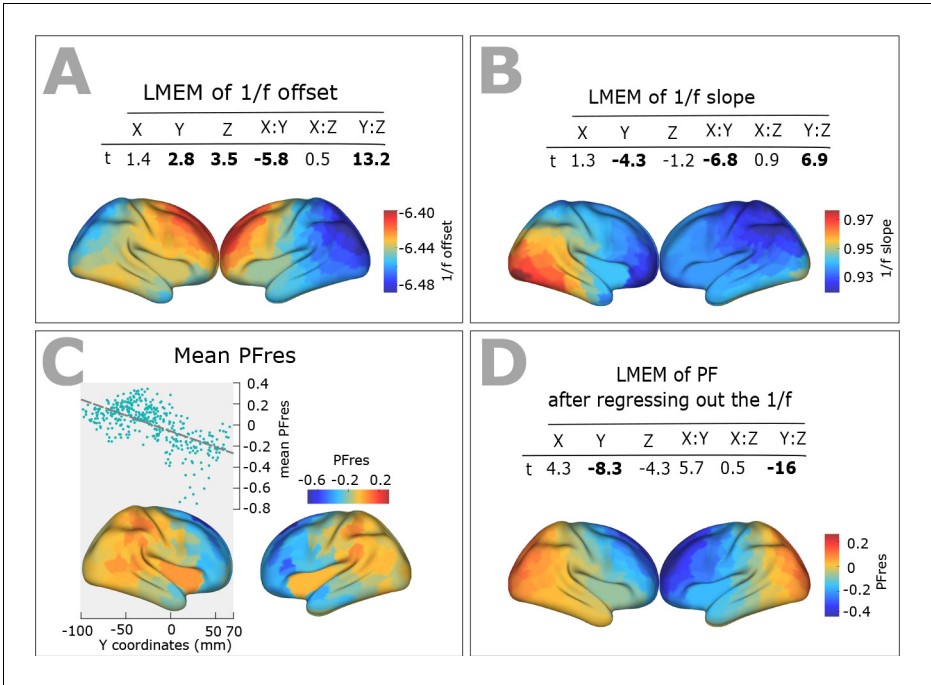

**Figure 2.** Spatial distribution of 1/f components (offset and slope) across human cortex. (**A**) Top panel: t-values obtained from linear mixed effect modeling of 1/f offset as a function of the coordinates of ROI centroids. Bottom panel: cortical map of the corresponding fixed effects. (**B**) LMEM was applied on 1/f slope, analogous to the 1/f offset. The slope and offset of 1/f component were estimated for each ROI and participant, using the FOOOF package (see Materials and methods section for further details). (**C**) Correlation between trimmed mean PFres (187 participants, 384 ROIs) and ROI's location along the y-axis (posterior to anterior) (r = −0.63, p<<0.001). The residual PF scores (PFres) were obtained after regressing out the contribution of 1/f offset and slope values (fixed effect) from PF, using LMEM. (**D**) t-values obtained from linear mixed effect modeling of PF as a function of the coordinates of the ROI centroids (LMEM; t-values: Y = −8.3, Z = −4.3, Y:Z = −16; all p<0.001). The cortical maps show the corresponding fixed effects.

from posterior to anterior regions (t-values: Y = 49.7, Z = −29.3, Y:Z = 16.2; all p<<0.001). Notably, similar to the spatial gradients of PF, LMEM uncovered a secondary inferior-superior axis of CT. Having established the organizational axes of CT, we then tested for a significant relationship between CT and PF. LMEM (t = −13.8, p<<0.001) showed a significant negative relationship between PF and CT. Next, we asked the question if this relationship is still significant after removing from both, PF and CT, the effect of ROI coordinates (x,y,z). This was done by modeling the dependencies of PF and CT on ROI coordinates and computing the residuals PFres and CTres, respectively. These residuals describe individual spatial variations of PF and CT that cannot be explained by a linear model of their spatial location. PFres and CTres are still significantly related (LMEM: t = −6.9, p<<0.001, *Figure 3—figure supplement 1*) indicating that they are more directly related to each other than can be explained by their individual dependency on location (x,y,z). This result suggests that peak frequency is related to structural features that likely represent cortical hierarchies.

We further tested the relationship between PF gradients and cortical hierarchies along the anatomically defined and well-established visual hierarchy. Following an approach by *Michalareas et al., 2016*, we selected seven cortical regions showing strong homology to macaques visual areas (V1, V2, V4, MT, DP, TEO, 7A) using the cortical parcellation of *Glasser et al., 2016*. *Figure 4A*, top panel displays a schematic representation of the seven regions. We modeled spatial changes of PF along the visual hierarchy, using LMEM (see Materials and method section for details), and found a significant decrease of PF (t = −10.1, p<<0.001). A similar analysis was also performed on CT and found a significant increase of CT along the visual hierarchy (t = 54.9, p<<0.001).

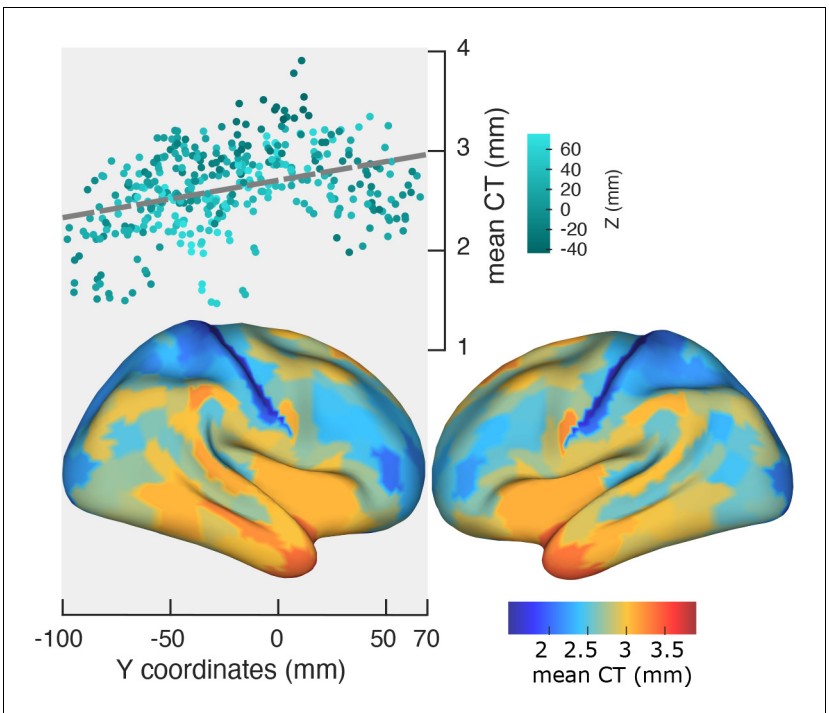

**Figure 3.** Spatial gradient of cortical thickness along the posterior-to-anterior direction. Top panel: correlation between mean cortical thickness and ROI's location along the y-axis (posterior to anterior) (r = −0.84, p<<0.001). Bottom panel: cortical map of trimmed mean PF across 384 cortical ROIs.

The online version of this article includes the following figure supplement(s) for figure 3:

**Figure supplement 1.** Relationship between PF (187 participants, 384 ROIs) and CT, after regressing out the effect of ROI coordinates.

**Figure supplement 2.** Relationship between PF (187 participants, 384 ROIs) and CT.

---

*Figure 4A*, bottom panel (bars) shows the fixed effects of the LMEM applied separately on PF and CT scores.

Previous studies have shown that cortical regions can be contextualized in terms of eight canonical resting-state networks comprising three sensory (visual, somatosensory, and auditory) and five higher order association networks (frontoparietal, cingulo-opercular, default mode, dorsal attention, and ventral attention; *Figure 4B*; *Ito et al., 2017*). Markers of hierarchical microcircuit specialization such as the ratio of T1-weighted to T2-weighted MRI maps (T1w/T2w) are significantly different between sensory and association areas (*Burt et al., 2018*; *Demirtaş et al., 2019*). Here, we extended this approach to our measures to test for differences in PF/CT between sensory and association networks. Following *Ito et al., 2017*, we assigned all cortical regions to eight resting-state networks and defined a categorical variable comprising eight labels corresponding to the networks. This categorical variable was used as a fixed effect (independent variable) and the PF/CT as a response variable for an LMEM, to test the effect of networks on cortical organization of PF/CT. The random effect was defined as in *equation1*. *Figure 4B* shows the effect of each network (fixed effect) for CT (top panel) and for PF (bottom panel). Error bars indicate the lower and upper bounds of LMEM for the fixed effect. Next, we applied ANOVA on LMEM and found a significant effect of resting-state networks for PF and CT (PF: F-stats = 264, p<<0.001; CT: F-stats = 746, p<<0.001). Similarly, PF and CT were significantly different between sensory and association areas (LMEM, PF: t = −11.1, p<<0.001; CT: t = 14.7, p<<0.001, *Figure 4B*). As expected, PF is higher in sensory areas compared to association areas while an opposite effect is observed for CT. These results are largely caused by the fact that networks differ in their location and that higher order associative brain areas are located more anterior compared to sensory brain areas (most prominently in the visual domain). Still, we believe that the 'network representation' of our results is important to emphasize the point that these quantities (PF and CT) differ between networks. This is especially important for studies

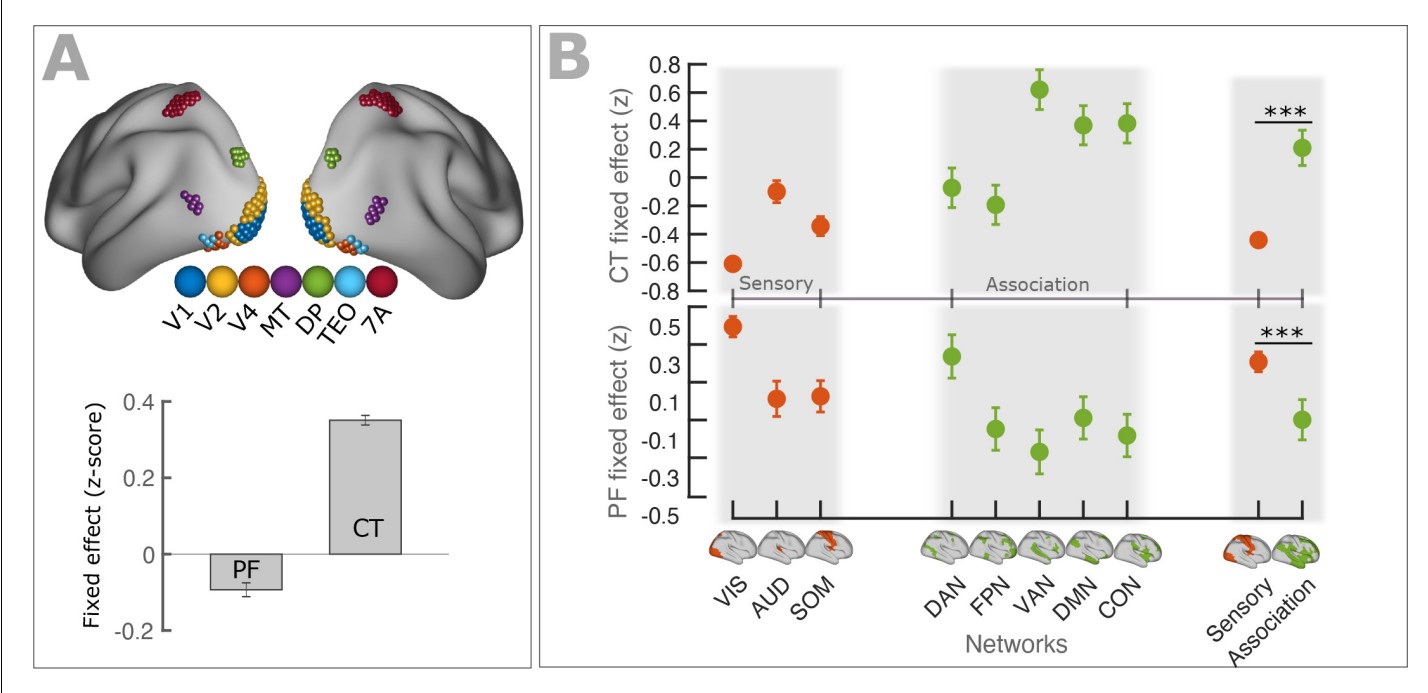

**Figure 4.** PF and CT variation along the cortex follows anatomical hierarchies. (**A**) Top panel: Schematic representation of seven regions (V1, V2, V4, MT, DP, TEO, 7A) used for defining visual hierarchy. Bottom panel: Each bar shows the fixed effect of the LMEM where the PF/CT was defined as a response variable and the visual hierarchy as an independent variable. We found a significant decrease of PF (t = −10.1, p<<0.001) but a significant increase of CT (t = 54.9, p<<0.001) along the visual hierarchy. To impose the hierarchical order of the seven ROIs in an LMEM, we defined a seven-element hierarchy vector for each participant and hemisphere (V = [1, 2, 3, . . ., 7]), whose elements refer to the hierarchical level of the corresponding ROI. The random effect was specified as in *Equation 1*. PF/CT values were standardized before LMEM analysis. This model tests the significance of PF/CT changes along the specified hierarchy. (**B**) Fixed effect per network obtained from linear mixed effect modeling of CT (top panel) and PF (bottom panel) as a function of networks (independent variable), where networks were specified as a categorical variable. The random structure was defined as in *Equation 1*. Fixed effect per network indicates the effect of that network on PF/CT. The network variable was defined as a categorical variable by assigning cortical regions to eight functional resting-state networks comprising three sensory ('VIS', visual; 'AUD', auditory; and 'SOM', somatomotor) and five association ('DAN', dorsal 670 attention; 'FPN', frontoparietal; 'VAN', ventral attention; 'DMN', default mode; and 'CON', cingulo-opercular) networks. We applied ANOVA on LMEM fit and computed F-stat for the fixed effect. (PF: F-stats = 264, p<<0.001; CT: F-stats = 746, p<<0.001). PF values were significantly lower in association RSNs (except for DAN) than in sensory RSNs (t = −11.1, p<<0.001), whereas CT values were significantly higher in association RSNs than in sensory RSNs (t = 14.1, p<<0.001). Error bar indicates the lower and upper bounds of LMEM for the fixed effect. The online version of this article includes the following figure supplement(s) for figure 4:

**Figure supplement 1.** Distribution of the location independent PF (PFres) among resting state networks.
**Figure supplement 2.** Distribution of location independent CT (CTres) among resting state networks.

focusing a-priori on these anatomically defined networks and where these differences may confound other results. However, we also tested if PF and CT differ between networks irrespective of their location. This was done by removing from both measures changes that can be explained by linear dependencies on x,y,z. Interestingly, after regressing out the effect of spatial location, a significant difference among networks remained for PF (*Figure 4—figure supplement 1*) and CT (*Figure 4—figure supplement 2*). This was the case when looking at the standard resting-state networks and also when testing sensory against association areas. These results indicate that, beyond a global effect of location, networks still differ significantly in PF and CT after removing the linear effects of location.

## Spatial gradients are frequency specific

In the results presented so far, we defined the PF per ROI as the most prominent peak in the spectrum. Some ROIs, however, showed more than a single spectral peak possibly indicating the presence of several gradients. We tested this in four frequency bands derived from our data. *Figure 5* shows a histogram (across ROIs and participants) of all detected spectral peaks. Interestingly, this

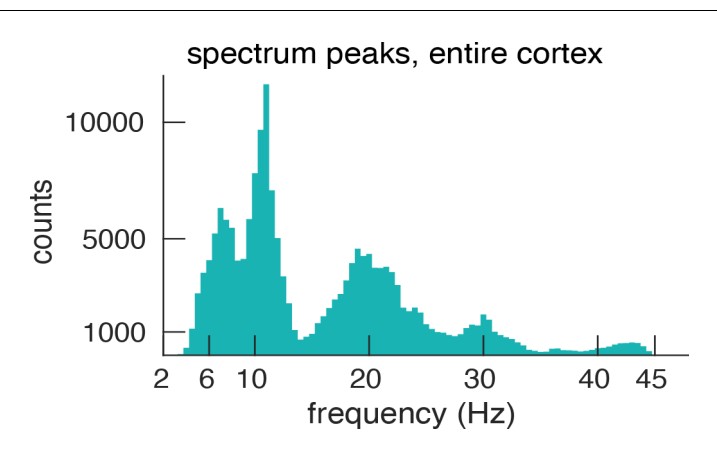

**Figure 5.** Histogram of spectral peaks. Histogram of all detected spectral peaks (across ROIs and participants) delineates the classical frequency bands used in the EEG and MEG literature (theta 3.5–7.5 Hz, alpha 8.5–13 Hz, low-beta 15–25 Hz and high-beta 27.5–34).

histogram of peak frequencies clearly delineates the classical frequency bands that are traditionally used in the EEG and MEG literature – but they are here derived in a purely data-driven manner (4–7.5 Hz (theta), 8.5–13 Hz (alpha), 15–25 Hz (low beta) 27.5–34 Hz (high beta), for subsequent analysis both beta bands were combined).

We identified the band-specific PF as the frequency of the peak with largest amplitude in a given frequency band for each ROI and participant. Analogous to the PF analysis, we used LMEM to model band-specific PF as a function of the ROIs' coordinates (*Figure 6*). We found a significant decrease of alpha peak frequency (correlation with y-axis: r = - 0.87, p<<0.001; LMEM: Y, t = −10, p<<0.001; =Y:Z, t = 3.2, p=0.001, *Figure 6B and E*) only along the posterior-anterior direction, while theta- (correlation with y-axis: r = 0.40, p<0.001; LMEM: Y, t = 7.4, p<<0.0.001; Z, t = −7, p<<0.001; Y:Z, t = −8, p<<0.001, *Figure 6A and D*) and beta-range (correlation with y-axis: r = 0.9, p<<0.001; LMEM: Y, t = 9.3, p<<0.001; Z, t = 4.7, p<<0.001; Y:Z, t = 2.3, p<<0.001, *Figure 6C and F*) frequencies significantly increased along the same direction. Moreover, LMEM identified a secondary significant gradient of theta- and beta-PF along the z-axis.

This spatial gradient of band-specific PFs was independent of spatial changes of 1/f slope and offset (*Figure 6—figure supplement 1*) but significantly correlated with that of CT scores (*Figure 6—figure supplement 2*).

## Discussion

This study is the first comprehensive demonstration of frequency gradients across the human cortex using a large set of resting-state MEG recordings. We found that the dominant peak frequency in a brain area decreases significantly, gradually and robustly along the posterior-anterior axis, following the global cortical hierarchy from early sensory to higher order areas. This finding establishes a frequency gradient of resting-state brain rhythms that complements previous anatomical studies reporting a posterior-anterior gradient in microscale and macroscale anatomical features of animal and human cortex (*Huntenburg et al., 2018*).

Several MEG/EEG studies have demonstrated that alpha activity (~10 Hz) is strongest in occipito-parietal brain areas and theta activity (~5 Hz) strongest in more frontal brain areas (*Chiang et al., 2011*; *Voytek et al., 2010*). There is also evidence that the dominant frequencies differ between brain areas (*Groppe et al., 2013*; *Hillebrand et al., 2016*; *Keitel and Gross, 2016*). However, we provide the first comprehensive (in frequency and space) statistical model of frequency gradients in a large resting-state brain activity.

Our results are consistent with a recent invasive study showing a systematic decrease of peak frequency from posterior to anterior brain areas in ECoG recordings of epilepsy patients (*Zhang et al., 2018*).

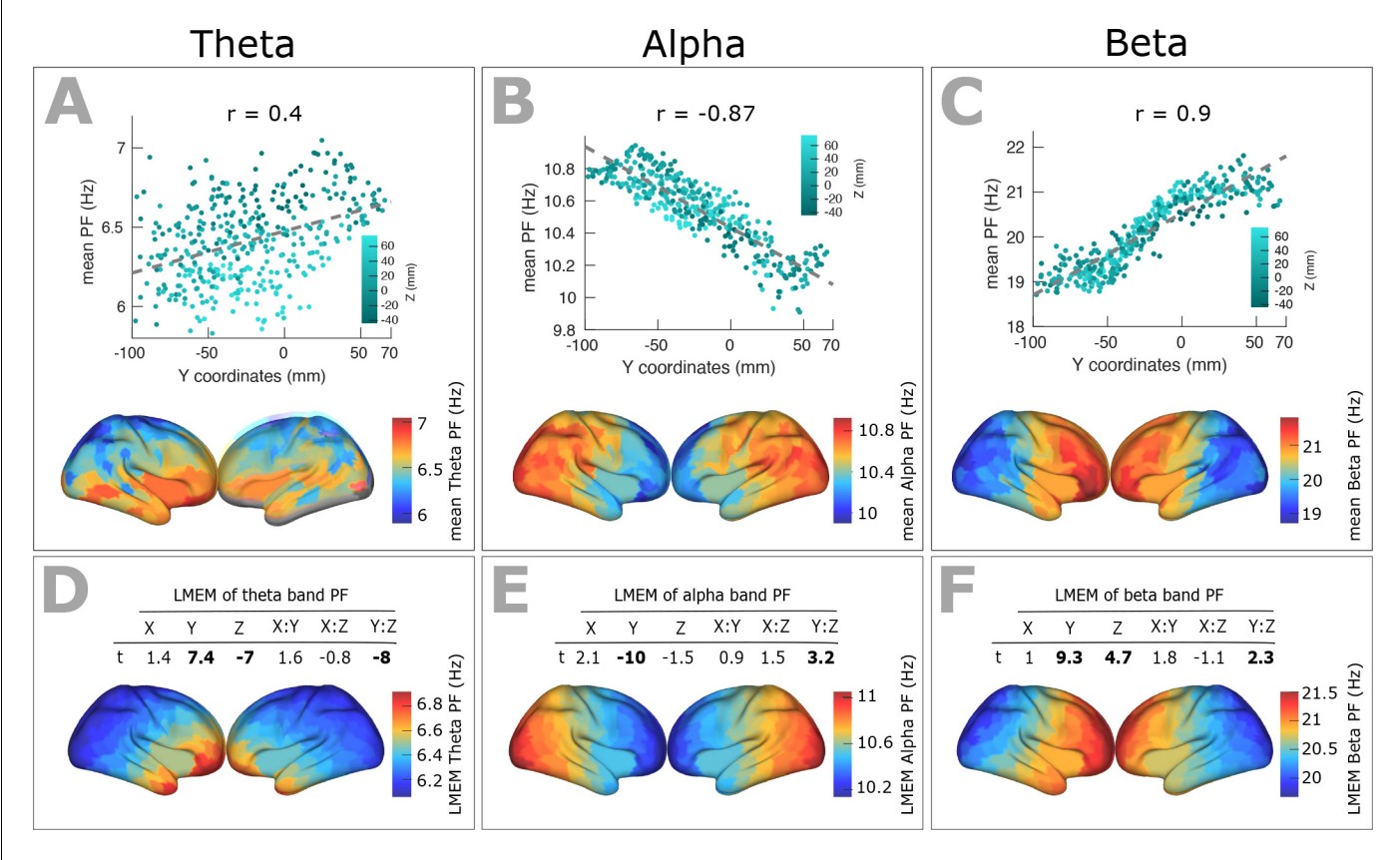

**Figure 6.** Spatial gradients of band-specific PFs across human cortex follows the posterior-anterior direction. (A, B, and C) Top panel: Dependency between the trimmed mean of band-specific PF (187 participants, 384 ROIs) and the ROI's location along the y-axis (posterior to anterior) for theta (r = 0.4, p<0.001), alpha (r = −0.87, p<0.001), and beta (r = 0.9, p<0.001) bands, respectively. Points are colored according to their Z coordinates. Bottom panel: distribution of trimmed mean band-specific PFs across 384 cortical ROIs. (D, E, and F) Top panel: t-values obtained from linear mixed effect modeling of band-specific PF (theta, alpha, and beta bands, respectively) as a function of the coordinates of the ROI centroids. Bottom panel: cortical map of the corresponding fixed effect parameters (see *Equation 2* for details).

The online version of this article includes the following figure supplement(s) for figure 6:

**Figure supplement 1.** Spatial gradients of band-specific PF across the cortex after factoring out the impact of 1/f components (offset and slope).

**Figure supplement 2.** Dependency between band-specific PF and CT.

A differentiating feature of our approach was that we used a large number of healthy participants (N = 187), reconstructed cortical activity from noninvasive MEG recordings, and considered further anatomical features (i.e. cortical thickness). Notably, estimating the power spectrum in finely parcellated ROIs allowed us an accurate and robust identification of peak frequencies and characterization of their spatial gradients across the entire cortical surface. Importantly, we focus on peaks in the power spectrum that indicate the presence of rhythmicity in the neuronal activity, instead of focusing on predefined frequency bands where these rhythms might be absent. As slope and offset of frequency gradient could dramatically vary across participants, averaging across participants may not yield a reliable representation of PF gradient. Instead, we used mixed effect modelling of PF along the cortical hierarchies, where the between-participant variability was taken into account as a random effect. For example, the peak frequency in occipital brain areas and the slope of the frequency gradient varies across individuals, and thus the intercept and slope of the corresponding least square fit change. These individual differences are specifically modelled by a mixed effect model as random effects. Importantly, a mixed effect model will only show a significant gradient if it is significant across cortical areas at the individual level, and consistent across participants at the group level.

Results of our analyses showed that, just as peak frequency significantly decreased along the posterior-anterior axis, CT significantly increased in the same direction, which resulted in a significant anticorrelation between PF and CT. The observed correlation holds after removing the effect of spatial location (x,y,z). This seems to indicate that PF and CT are more closely related to each other than can be explained by spatial location alone. Since cortical hierarchies do not strictly follow a single linear trajectory in space (e.g. posterior-anterior) our results are consistent with the idea that both PF and CT, follow cortical hierarchies. Indeed, such local spatial gradients have been reported in multiple features of cortex during auditory perception (*Jasmin et al., 2019*) and visual processing streams (*Himberger et al., 2018*). From a broader view, local gradients could mirror complex organization of gradients in human cortex and support the approach of global gradient along the sensory to transmodal areas (*Huntenburg et al., 2018*). On the other hand, posterior-anterior gradient of PF was significant after subtracting CT scores from PF values. This suggests a partial independence of both measures. Since PF is a measure derived from brain activity the reported gradient could be modulated dynamically depending on cognitive state or task demands. Further studies are needed to investigate this in more detail.

We further addressed the question if our results can be explained by the linear superposition of activity from an occipital alpha source and a frontal theta source. Along the posterior-anterior axis, differential superposition of both sources could lead to a frequency gradient, due to imperfect unmixing of the signals. However, our analysis revealed that a significant frequency gradient is already evident within 2–3 cm of V1 where the effect of a frontal theta source (which has on average a lower power compared to occipital alpha) is negligible. Furthermore, A more comprehensive analysis showed existence of this gradient for posterio-pariatal, central, and frontal areas, separately, acknowledging the globally and continuously decreasing nature of the posterior-anterior gradient.

Additional supporting evidence can be drawn from intracranial studies, where the data is directly recorded from cortex. *Zhang et al., 2018* have shown that oscillations generally propagate in a posterior-to-anterior direction because they are coordinated by an overall decrease in intrinsic oscillation frequency from posterior to anterior regions (see Figures S6 and 7 of *Zhang et al., 2018*). Overall, this indicates the existence of a gradual decrease of PF along the posterior-anterior axis.

What is the potential functional role of this frequency gradient? Zhang et al. demonstrated the existence of travelling waves along the frequency gradient (*Zhang et al., 2018*). Interestingly, they found that local frequencies along the posterior-to-anterior direction are positively correlated with waves' propagation speed and direction consistent with a proposed model of travelling waves based on weakly coupled oscillators (WCO) (*Ermentrout and Kleinfeld, 2001*). Travelling waves along the posterior-anterior axis have also been reported during visual stimulation (*Alamia and VanRullen, 2019*; *Lozano-Soldevilla and VanRullen, 2019*). These travelling waves might serve to drive neural communication along the cortical hierarchy possibly through nested gamma oscillations thereby linking travelling waves to the concept of pulsed inhibition (*Bahramisharif et al., 2013*). In addition, travelling waves have been associated with memory consolidation and learning (*Muller et al., 2018*). It is of interest to note that frequency gradients have been reported previously in the entorhinal cortex (*Giocomo et al., 2011*; *Giocomo and Hasselmo, 2009*). Here, a frequency decrease and corresponding travelling waves have been observed in the dorsal-ventral direction and have been related to a representational gradient of spatial scales from coarse to fine (*Muller et al., 2018*). Indeed, converging evidence across recording methods, species and cortical domains suggests that representations become more 'integrated' with decreasing 'dominant' frequency of the underlying neuronal population. A prime example is the auditory cortex where response latencies and complexity of processing increase along the posterior-anterior axis (*Jasmin et al., 2019*). This is also mirrored by an increase in cortical thickness and increased ratio of feedback to feedforward connections along this axis. Similar observations have been made across more widely distributed cortical areas where timescales of intrinsic fluctuations in spiking activity increase from posterior to anterior brain areas (*Murray et al., 2014*). Not surprisingly, these time scales are largely determined by the time constants of synaptic transmission (*Duarte et al., 2017*). But interestingly, in a computational model of activity in macaque cortex using anatomical connectivity a gradient of time scales also emerges with short, transient responses to input in sensory areas and slower, sustained responses in higher order areas (*Chaudhuri et al., 2015*) (see also *Kiebel et al., 2008*).

Our approach additionally revealed that cortical peak frequencies decrease systematically along the inferior-superior axis. As seen in *Figure 1* this seems to result from the fact that higher order

frontal areas with lower PF have higher z-coordinates compared to the early sensory areas with higher PF. A similar inferior-superior gradient was also observed for cortical thickness. This finding is supported by a recent study reporting an inferior-superior gradient organization of the CT in human and macaque monkeys (*Valk et al., 2020*), and has been attributed to the organization patterns expected based on the theory of dual origin (*Goulas et al., 2018*).

Our detailed analysis was based on the cortical ROIs' spectral peak with strongest power (PF). However, we identified all peaks in the power spectrum of each ROI. Since spectral peaks indicate the presence of brain rhythms, this data represents a comprehensive overview of these rhythms across the cortex. The histogram of spectral peaks across ROIs and participants provided a data-driven definition of frequency bands. Interestingly, the histogram delineates the classical frequency bands with histogram peaks centering at 4–7.5 (theta), 8.5–13 (alpha), 15–25 (low-beta) 27.5–34 (high-beta) (see *Figure 5*). This is the first MEG study to our knowledge to identify frequency bands from peak frequencies in a large data set (see *Groppe et al., 2013* for a similar approach in a smaller sample of ECoG data).

We further analyzed these specific frequency bands for gradients and found significant posterior-anterior frequency changes in the theta, alpha and beta frequency band. Results in the alpha band mirrored the previous results based on the overall strongest peak frequency. Interestingly, and in contrast to the alpha band, peak frequencies increased along the posterior-anterior direction in the theta and beta frequency band. In the model used by Zhang et al. this would correspond to travelling waves from anterior to posterior brain areas (*Zhang et al., 2018*) that might represent frequency channels for top-down effects (*Michalareas et al., 2016*; *Wang, 2010*).

An alternative explanation for the observed posterior-to-anterior or anterior-to-posterior changes of the band-specific PFs may come from the laminar organization of the cortex, where several layers exhibit distinct frequency profiles (*Bastos et al., 2018*) and thickness patterns (*Wagstyl et al., 2020*). In particular, similar to our results for the spatial gradients of band-specific PFs, Wagstyl and colleagues (*Wagstyl et al., 2020*) have identified both increasing and decreasing gradients of thickness along the posterior-anterior axis for cortical layers, in the somatosensory, auditory, and motor cortex (see Figure 6 of *Wagstyl et al., 2020*).

In summary, our findings show that peak frequencies of cortical areas form a spatial gradient, which follows the global posterior-anterior hierarchy as well as local anatomical hierarchies. Previous research also points to spatial gradients in multiple features of the human and animal cortex. Further research might explore implications of frequency gradients in different cognitive states, disease, and aging.

## Materials and methods

### Experimental design

In this study, we used the MOUS dataset (*Schoffelen et al., 2019*) (https://data.donders.ru.nl/collections/di/dccn/DSC_3011020.09_236?1) which, among others, contains five minutes of resting state MEG recordings that is available for 197 out of a total of 204 healthy participants (age: mean = 22, range = 18–32, gender: 94 females). The participants were instructed to think of nothing specific while focusing on the fixation cross at the center of the screen. Data was collected using a CTF 275-channel radial gradiometer system, and sampled at 1200 Hz (0–300 Hz bandpass), and additional 29 reference channels for noise cancellation purposes.

The anatomical images of the head were obtained with a SIEMENS Trio 3T scanner using a T1-weighted magnetization-prepared rapid gradient-echo (MP-RAGE) pulse sequence, with the following parameters: volume TR = 2300 ms, TE = 3.03 ms, eight degree flip-angle, one slab, slice-matrix size = 256 × 256, slice thickness = 1 mm, field of view = 256 mm, isotropic voxel-size = 1.0 × 1.0×1.0 mm.

After removing 10 participants containing excessive ocular-, muscular-, and cardiac-related artefacts, or lacking any visible spectral peak in source space, we used 187 participants for our analyses.

### MEG preprocessing

All preprocessing analyses were performed using the Fieldtrip package (*Oostenveld et al., 2011*). Gradiometer signals were converted to synthetic third-order gradients, high-pass filtered at 0.5 Hz,

and low-pass filtered at 140 Hz (Butterworth, 4th order). Line noise was rejected using a DFT filter at 50 and 100 Hz. After down-sampling the data to 300 Hz, outlier channels/time segments were rejected using visual inspection of their time course, spectrum and topography. Next, we used independent component analysis (ICA) to identify and remove signal components related to eye blinks/movements and cardiac activity. To this end, we performed ICA, using the infomax algorithm (*Bell and Sejnowski, 1995*), on a 30-dimensional signal subspace, for computational efficiency. ICs related to artifacts were identified based on their spatial topography and signal time course, and the identified spatial topographies were projected out of the sensor data. This resulted in 3.7 components on average to be rejected (range 1–6).

## MRI analysis

From T1-weighted anatomical images of participants, brain/skull boundary and cortical surfaces (white matter and pial matter) were generated using SPM (*Penny et al., 2011*) and Freesurfer (version 5.1)(http://surfer.nmr.mgh.harvard.edu). The cortical surface was coregistered to a template with a surface-based coregistration approach (Caret software, http://brainvis.wustl.edu/wiki/index.php/Caret:Download), and downsampled to 8196 vertices (MNE software, martinos.org/mne/stable/index.html). Using the Caret software, the mid-thickness cortical surface (halfway between the pial and white matter surfaces) was generated. The cortex surface was parceled into 384 ROIs (192 per hemisphere) according to *Schoffelen et al., 2017*. 648 vertices located in the medial wall (sub-cortical areas) were excluded from further analysis.

The centroid of each parcel was specified as the vertex located at minimum geodesic distance from all other vertices of that parcel.

## Source reconstruction

Source reconstruction was performed using the linearly constrained minimum variance beamformer approach (*Van Veen et al., 1997*), where the lambda regularization parameter was set to 5%. This approach estimates a spatial filter for each location of a set of defined dipole locations (here: each of the 7548 non-midline vertices of the mid-thickness cortical mesh), based on the forward model of that location and the sensor covariance matrix. The forward model was computed using the 'single-shell' method, with the brain/skull boundary as volume conduction model of the head. The sensor covariance matrix was computed between all MEG-sensor pairs, as the average covariance matrix across the 2 s time window covariance estimates.

## ROI spectrum

For each anatomical ROI, we performed dimensionality reduction using singular value decomposition (svd) on all vertex timeseries. We retained the required number of components to account for 95% of the variance for each ROI (typically 15 components). Component time courses were segmented to 2-s epochs, from which power spectra were computed using a multitapered Fast Fourier transform, using discrete prolate spheroidal sequences (dpss) as windowing function, with 2 Hz spectral smoothing. To obtain a single spectrum for each ROI (ROI spectrum), we pooled spectra of epochs across components and computed the 10% trimmed mean across them. Averaging after leaving out 10% of data from left and right tails of the spectra distribution offers a more robust estimate.

## Peak frequency (PF) detection

We estimated 1/f component of spectrum between 3 and 45 Hz using the FOOOF algorithm (*Haller et al., 2018*). The algorithm fits a linear approximation of 1/f in log-log spectrum and computes the corresponding slope and offset parameters. Next, we subtracted the estimated 1/f component from the spectrum to obtain a 1/f corrected spectrum per ROI. To identify spectral peaks, we used the MATLAB 'findpeaks' function. We extracted all peaks but most of the analysis is based on the peak frequency with the strongest power in the original spectrum that includes the 1/f background.

## Cortical thickness (CT)

We used the Freesurfer package to obtain estimates of CT scores. The CT value of a vertex was computed as the distance between corresponding white matter and pial surface vertices. To obtain thickness values of a ROI, we averaged CT across the vertices of that ROI.

## Statistical analysis

As described above, we computed PF values for 384 ROIs (197 ROIs per hemisphere) of 187 participants. In our statistical analyses, we aimed to investigate the spatial/hierarchical organization of PF across the human cortex, but also control for the between-participant variability. To meet this purpose, we used linear mixed effect modeling (LMEM). The distinctive feature of LMEMs is that a response variable is modeled as a linear combination of (1) population characteristics that are assumed to be shared by all individuals (fixed effects), and (2) participant-specific effects, that are unique to a particular individual in the population (random effects).

To investigate the spatial organization of PFs across the cortex, we specified the PF as response variable and the coordinates of ROI centroids (X: left to right, Y: posterior to anterior, and Z: inferior to superior) plus their two-way interactions (XY, XZ, YX) as fixed effects. The inclusion of two-way interaction as predictors allows the model to adapt well to the cortex geometry. As our random structure, we nested the PFs within participants as well as within hemispheres to account for the variability between participants and hemispheres. *Equation 1* shows the specified LMEM

$$PF_j = \beta_0 + S_{0j} + (\beta_1 + S_{1j})X + (\beta_2 + S_{2j})Y + (\beta_3 + S_{3j})Z + \beta_4 XY + \beta_5 XZ + \beta_6 YZ + e_j \qquad (1)$$

where the response variable PF for the participant $j$ is related to baseline level via ($\beta_0$), to ROI centroids (fixed effects) via ($\beta_i, i \in \{1,2,...,6\}$), and to error ($e_j \sim N(0, \sigma^2)$). To address the variation of predictors for participant $j$, we specified both random intercepts ($S_{0j}$) and slopes ($S_{ij}, i \in \{1,2,3\}$) for random effects. For the sake of model simplicity, no random effect was specified for two-way interactions. We estimated the fixed effect predictions for a ROI located at centroid coordinates of ($x, y, z$) as follows

$$PF_{xyz} = \beta_0 + \beta_1 x + \beta_2 y + \beta_3 z + \beta_4 xy + \beta_5 xz + \beta_6 yz \qquad (2)$$

In our analysis, we included only significant predictors for *Equation 2*. We used an analogous approach, to test the significance of spatial changes of CT and 1/f parameters across the cortex.

To examine if the spatial distribution of PF across the cortex is independent of the spatial changes of 1/f parameters (slope and offset), we fitted a LMEM, where we set the PF as a response variable, the 1/f slope and offset scores as fixed effects, and the between- participant and hemisphere as random effects. Prior to LMEM, we standardized the PF, 1/f slope and offset scores for each participant by subtracting mean and dividing by standard deviation (z-score). Next, we estimated the coefficients for the fixed effects (1/f parameters) and regressed them out to obtain the residual PF (PFres) scores, which reflect a subspace of PF that cannot be explained by 1/f parameters. We again modeled the obtained PFres scores as a function of ROI centroids as described above (see *Equation 1*).

To obtain the correlation between PF and CT scores, we initially computed the 10% trimmed mean across participants for each ROI and performed a robust correlation (skipped Pearson correlation, Robust Correlation Toolbox) (*Pernet et al., 2012*) between the trimmed mean values. To address the between-participant variability, we first standardized PF and CT scores (as described above), and conducted LMEM, where we specified the PF as response variable and the CT as a fixed effect predictor. The random effect was set according to *Equation 1*. Moreover, we aimed to obtain a correlation value between PF and CT that is independent of spatial location. We first applied LMEM separately for PF and CT, modeling each as a function of ROI coordinates (see *Equation 1*), and computed the corresponding residuals (PFres and CTres) for each ROI and participant. Subsequently, we applied LMEM between PFres and CTres values (analogous to PF and CT).

To test for the significance of PF changes along the established visual hierarchy comprising seven regions (V1, V2, V4, MT, DP, TEO, 7A), chosen according to *Michalareas et al., 2016*, we used an LMEM. To impose the hierarchical order of those seven ROIs in our LMEM, we defined a seven-element hierarchy vector for each participant and hemisphere (V = [1, 2, 3, . . ., 7]) whose elements refer

to the hierarchical level of the corresponding ROI. The random effect was specified as in *Equation 1*. PF values were standardized before LMEM analysis. This model tests the significance of PF changes along the specified hierarchy. An analogous analysis was applied to CT scores of those seven ROIs.

To statistically assess the effect of eight resting-state networks on PF, we used a recently released multi-modal parcellation of the human cortex (*Glasser et al., 2016*) and assigned cortical regions to eight functional resting-state networks comprising three sensory ('VIS', visual; 'AUD', auditory; and 'SOM', somatomotor) and five association ('DAN', dorsal 670 attention; 'FPN', frontoparietal; 'VAN', ventral attention; 'DMN', default mode; and 'CON', cingulo-opercular) networks (*Ito et al., 2017*). Next, we used an LMEM analysis, where we specified the PF per region and individual as a response variable, and the assigned label for each region (({'VIS', 'AUD', 'SOM', 'DAN', 'FPN', 'VAN', 'DMN', 'CON'}) as an independent (categorical) variable. The random structure was defined as in *Equation 1*. This analysis obtains t-values for each network representing the significant effect of that network on PF. Subsequently, we applied ANOVA on LMEM fit and computed F-stat for the fixed effect. A similar analysis was performed to test the effect of resting-state networks on CT scores.

All statistical analyses were conducted in Matlab version 9.5 (R2018b). We used the 'fitlme' function to perform the LMEM analysis.

## Acknowledgements

The authors thank Peter Hagoort, who initiated and financed the MOUS project, as well as the MOUS-team that collected, and made publicly available, the data used for this study. JG was supported by Interdisciplinary Center for Clinical Research (IZKF) of the medical faculty of Münster (Gro3/001/19) and by the DFG (GR 2024/5-1).

## Additional information

### Funding

| Funder | Grant reference number | Author |
|---|---|---|
| University of Münster | | Keyvan Mahjoory |
| Nederlandse Organisatie voor Wetenschappelijk Onderzoek | 864.14.011 | Jan-Mathijs Schoffelen |
| IZKF | Gro3/001/19 | Joachim Gross |
| Deutsche Forschungsgemeinschaft | GR 2024/5-1 | Joachim Gross |

The funders had no role in study design, data collection and interpretation, or the decision to submit the work for publication.

### Author contributions

Keyvan Mahjoory, Resources, Software, Formal analysis, Validation, Investigation, Visualization, Methodology, Writing - original draft, Writing - review and editing; Jan-Mathijs Schoffelen, Data curation, Software, Formal analysis, Methodology, Writing - review and editing; Anne Keitel, Writing - review and editing; Joachim Gross, Supervision, Funding acquisition, Validation, Methodology, Writing - review and editing

### Author ORCIDs

Keyvan Mahjoory (iD) https://orcid.org/0000-0002-0386-1135
Jan-Mathijs Schoffelen (iD) https://orcid.org/0000-0003-0923-6610
Anne Keitel (iD) http://orcid.org/0000-0003-4498-0146
Joachim Gross (iD) https://orcid.org/0000-0002-3994-1006

### Decision letter and Author response

Decision letter https://doi.org/10.7554/eLife.53715.sa1
Author response https://doi.org/10.7554/eLife.53715.sa2

## Additional files

### Supplementary files
• Transparent reporting form

### Data availability
We used a publicly available dataset for this study (https://data.donders.ru.nl/collections/di/dccn/DSC_3011020.09_236?0).

The following datasets were generated:

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
