## [Decision Letter]

**Acceptance summary:**

The manuscript makes an important contribution to the current literature showing how the frequency gradients of human resting-state neuronal oscillations are organized in accordance with cortical hierarchy.

**Decision letter after peer review:**

[Editors’ note: the authors submitted for reconsideration following the decision after peer review. What follows is the decision letter after the first round of review.]

Thank you for submitting your work entitled "The frequency gradient of human resting-state brain oscillations follows cortical hierarchies" for consideration by *eLife*. Your article has been reviewed by three peer reviewers, including Laura Dugué as the Reviewing Editor and Reviewer #1, and the evaluation has been overseen by a Senior Editor.

Our decision has been reached after consultation between the reviewers. Based on these discussions and the individual reviews below, we regret to inform you that your work will not be considered further for publication in *eLife*.

This manuscript presents results from a large dataset of human resting-state MEG recordings. The authors investigated the timely topic of the spatial distribution of low frequency brain oscillations, and found an anterior-posterior gradient in the frequency of brain oscillations. They further show a gradient in cortical thickness (CT), and suggest that CT correlates with oscillations frequency. Although investigating the spatial distribution of brain oscillations goes beyond simply studying their temporal dynamics, other studies have shown before that oscillation frequency varies across the brain. Reviewers have also noticed that such spatial gradients are in fact more specific to the alpha frequency. Some methodological aspects have additionally been raised including addressing the question of spatial leakage, and the possible confound of inter individual differences. Based on these comments and others described below, and on following discussions, the three reviewers concluded that this manuscript will not be considered further for publication in *eLife*.

Reviewer #1:

Mahjoory et al. present new results on a publicly available, resting-state MEG recording dataset from a large pool of participants. This well-written manuscript provides a systematic analysis of the spatial distribution of high-power, low-frequency, oscillations, based on non-invasive recordings of healthy, human participants. The manuscript is interesting in that the authors go beyond the mere temporal characterization of neural activity, using advanced source reconstruction approach and linear mixed effect modeling for statistics. Based on their results, the authors report a posterior-anterior spatial gradient of the dominant peak frequency, and further our understanding of the link between structure and function.

Numbered summary of any substantive concerns

1) My main concern regards the authors' statement that there is a decrease of the intrinsic resonance frequency from early sensory to higher-order area.

i) In their analysis, the authors identify PF on the 1/f-free power spectra and then select the PF that has the highest power based on the original power spectra. Isn't this approach necessarily increasing the probability to detect a pic at low (around alpha) frequencies? In other words, such decrease in oscillations' frequency is actually a decrease of, specifically, alpha oscillations. The band-specific PF analysis actually suggests such conclusion.

ii) It has been shown before (e.g. Rosanova et al., 2009) that different lobes of the brain have different "intrinsic resonance frequencies," e.g. beta frequency appears in anterior regions, and the proposed analysis does not seem sensitive to it. The Band-specific PF analysis, however, shows that indeed beta increases from posterior to anterior regions.

iii) Based on the results reported in Figure 4A, the authors suggest that there is a change in PF and CT as a function of the region's function (i.e. sensory vs. associative regions). Yet, the data seem to suggest that the effect is driven by the location of the different regions: we see high PF in VIS and DAN, which are mainly located in the posterior part of the brain, and low PF in the other networks. And doesn't this interpretation fit better the fact that there is a link between CT and PF? In other words, change in CT are probably not sparse across the head as some of these networks seem to be, right?

2) Could the authors give an intuition of the variance in ROIs' size for a given participant and across participant, as well as how the size affects the power spectra?

3) The authors argue that there is a posterior-anterior gradient of CT, which strongly correlates with the PF gradient. From the results presented in Figure 3, it seems that the CT gradient is more strongly explained by the Z-axis (thicker in ventral than dorsal regions).

4) Could the authors give an intuition of how stable (within participant) these PF are over time (see for instance Haegens et al., 2014)?

5) Could the authors comment on the absence of gamma oscillations?

6) Could the authors comment on the generalizability of their results given the fact that they analyzed resting-state data?

Reviewer #2:

This study investigated a novel and interesting topic on whether there is anterior-posterior gradient in the frequency of brain oscillations using source-localized MEG data. It was found that the spatial gradient of strongest peak-frequency decreases gradually and robustly along the posterior-anterior axis following the hierarchy of early sensory and higher-order areas. The manuscript is themed around a robust spatial gradient of the frequency of brain oscillations while the main finding was the spatial gradient in the frequency of alpha oscillation but not across oscillatory frequency hierarchy. I would suggest reframing the text to be better in line the findings. Further, the main analysis are mostly sound but methodological details are missing and some parts are difficult to evaluate. Also the contribution of spatial leakage as confounding factor to the results has not been sufficiently well tested and should be better controlled for.

1) The strongest peak in the original power spectrum was selected. Taken the robust alpha oscillations in rest, these peaks were in the alpha-band range (8-12 Hz) as visible in Figure 1A. The results hence describe the spatial gradient of alpha oscillations in narrow frequency range of 7 to 12 Hz. Thus, I find the title of the manuscript "The frequency gradient of human resting-state brain oscillations follows cortical hierarchies" as well as structure of text somewhat misleading and suggestive of the presence of spatial oscillatory frequency hierarchy, which seems not to be the case. Also the spatial gradients of peaks in theta and beta frequencies were estimated, but these data are now only shown in the supplementary material, not thoroughly analysed and only discussed very superficially. It looks like these data were added only for the reviewers. I think the authors should decide do they want to present a frequency gradient or an alpha-frequency gradient and modify the manuscript accordingly.

2) The correlation of the frequency and cortical thickness was estimated to investigate if the frequency is correlated rather with cortical hierarchy than spatial location. A correlation of -0.14 was found (Figure 3B). Albeit very significant, the correlation itself is very weak as also shown by the large scattered peak frequencies. Compared to correlation of -0.84 of the peak frequency with posterior-anterior gradient, this result indicates that alpha peak frequency is not strongly related to cortical hierarchy. The Results and Discussion should be modified to emphasize this point. Moreover, the authors could perform a partial correlation analysis among the peak frequency, posterior-anterior direction and cortical thickness – does the correlation with thickness persist when the posterior-anterior gradient is factored out?

3) Related to the above, in Materials and methods the authors write that to obtain correlation between PF and CT scores, the robust correlation was performed (Pernet et al., 2012). Pernet et al. released a new open source Matlab toolbox for analysis of robust correlations using multiple statistical tests. Which of tests would the authors have used for the correlation analysis?

4) The observed spatial gradient of peak frequencies can reflect a true continuous frequency variation along an anatomical axis or, alternatively, it could be caused by source leakage of distributed alpha sources at different frequencies and different cortical areas. The authors argue that if the spatial gradient is due to leakage then there should not be frequency change in alpha oscillations in V1. As there was a significant correlation of PF of V1 and its nearby 0.5-1.5 cm sources, the authors concluded that there is no source-leakage from the frontal source that could explain the spatial gradient. However, the alpha sources that could become mixed and cause confounding the results do not need to be localized to V1 and frontal cortex and hence estimating the source-leakage of V1 alpha source appears inadequate in my opinion. Classical MEG alpha literature suggests that the strongest alpha sources could actually be located higher in the visual cortical hierarchy, in the parieto-occipital sulcus or in areas near precuneus which could well cause significant leakage affecting the analyses. Second, the spatial accuracy of source-localized MEG is > 1-2 cm and hence the area for which the source-leakage was estimated could have been too small to test whether there is leakage or not.

5) The authors have used a Beamforming approach for source reconstruction. Several details of the source reconstruction approach are missing or unclear. First, if Beamforming was used for the entire brain volume in voxel space, how was data transformed to cortical parcellations and how were the values for each parcel obtained? The sensor covariance matrix was computed for 2 sec trials. Which trials were these?

Reviewer #3:

Mahjoory et al. examined a large dataset of human MEG-recorded neuronal oscillations to probe the relation between oscillation properties across space and cortical thickness. Their main findings are that the brain shows anterior-posterior gradients in the frequency of neuronal oscillations and in the slope of 1/f nonoscillatory MEG patterns. They also find a matching A-P gradient in cortical thickness, and they suggest that cortical thickness correlates closely with oscillation frequency in individual subjects, even after accounting for mean anatomical variations in each pattern.

My enthusiasm for publishing this paper at *eLife* is limited because most of their findings are not especially novel. Specifically, it is already known that oscillation frequency varies across the brain (see work by Voytek et al., Groppe et al., Zhang et al., and others) and merely replicating this finding in a large open dataset is not extremely innovative. Similarly, Figure 5 on oscillatory peaks is not novel and neither is the result showing an overall A-P gradient in mean cortical thickness.

The paper's novel claim is showing that oscillation frequency correlates with cortical thickness even after accounting for mean anatomical patterns. However, this result is not adequate to justify publishing the entire paper, especially because the data underlying this result were not examined in much detail and there was no compelling and specific proposed mechanism to directly link these two phenomena. Further, I am concerned that this apparent correlation could actually be a reflection of intersubject differences in mean thickness and oscillation properties, rather than by a detailed region-by-region correspondence between these variables within-subject, as the authors suggest. Failing to rule out this possibility is a substantial weakness in this analysis and the authors could do more to demonstrate this effect at the within-subject level.

[Editors’ note: further revisions were suggested prior to acceptance, as described below.]

Thank you for submitting your article "The frequency gradient of human resting-state brain oscillations follows cortical hierarchies" for consideration by *eLife*. Your article has been reviewed by two peer reviewers, and the evaluation has been overseen by a Reviewing Editor and Laura Colgin as the Senior Editor. The following individual involved in review of your submission has agreed to reveal their identity: Satu Palva (Reviewer #2).

The reviewers have discussed the reviews with one another and the Reviewing Editor has drafted this decision to help you prepare a revised submission.

Summary:

The authors have revised the manuscript thoroughly based on the previous comments. The manuscript is a very nice addition to the current literature showing how the frequency gradients of human resting-state neuronal oscillations are organized in accordance with cortical hierarchy. Revisions are still required to clarify in the main text, the Abstract and the figures that the patterns are present within subject, as well as within frequencies in contrast to across frequencies.

Revisions:

1) A key part of the paper is showing frequency gradients both within and across subjects. The rebuttal letter does a good job explaining that the authors' statistical framework identifies this pattern robustly both within and across subjects. But the text related to this is still unclear. The authors should revise the text of the results to more clearly explain how their statistical framework identifies gradients both within and across subjects.

2) The authors should also consider revising their figures to show clear examples of within and across subject gradients. The scatter plots and brain plots in Figure 1, for example, are hard to understand because they seem to combine data both across subjects and regions. It would be very informative if the figures followed the statistical results.

3) In response to one of the reviewers, the revised paper now includes an analysis of analyses within specific bands. This new analysis is a bit hard to follow in the context of the paper because it is unclear how it relates to the paper's primary analyses. Is the idea that there are multiple oscillatory patterns at different frequencies that all show gradients simultaneously? Additional clarity here would be helpful.

---

## [Author Response]

[Editors’ note: The authors appealed the original decision. What follows is the authors’ response to the first round of review.]

This manuscript presents results from a large dataset of human resting-state MEG recordings. The authors investigated the timely topic of the spatial distribution of low frequency brain oscillations, and found an anterior-posterior gradient in the frequency of brain oscillations. They further show a gradient in cortical thickness (CT), and suggest that CT correlates with oscillations frequency. Although investigating the spatial distribution of brain oscillations goes beyond simply studying their temporal dynamics, other studies have shown before that oscillation frequency varies across the brain.

This is correct and we acknowledge that in our manuscript. However, this is not the main result of our study. The novel contributions of our study are the following: (1) We provide the first comprehensive (in space and frequency) statistical model of frequency gradients in a large resting-state data set. The main result is not that different brain areas feature oscillations of different frequencies but that the frequency changes systematically and globally along spatial (and hierarchical) gradients. The only study that had previously tested this hypothesis is Zhang et al., 2018. However, their result was based on a smaller sample, data was recorded from epilepsy patients with inherent pathological brain activity and the ECoG data was constrained by limited electrode coverage. In addition, only anterior-posterior changes were studied. We provide the first full 3D statistical model at the level of individual brain areas (showing for example a significant gradient in inferior to superior direction). (2) We provide the first full 3D statistical model of cortical thickness in a large data set and show that cortical thickness changes systematically in space and is correlated with peak frequency. (3) We show for the first time that frequency gradients follow cortical thickness (as a proxy of hierarchical level) more closely than can be explained purely by spatial location.

Reviewers have also noticed that such spatial gradients are in fact more specific to the alpha frequency.

We agree with the reviewers in that our framework for PF identification is more specific to low frequencies. In our approach, we determine peaks from the 1/f-free spectrum and define the PF as the peak with the largest amplitude in the original spectrum, which leads to the preference of low frequencies. We also had tested another approach, where a PF was defined as a peak with the largest amplitude in the 1/f-free spectrum, thereby, the resulting PFs were in a wide range covering both alpha and beta rhythms. However, we found the PFs obtained from the latter approach extremely dependent on goodness of the 1/f fit, lacking a systematic pattern, and heavily inconsistent across participants. To address reviewers’ perfectly valid comment, we reframed the manuscript and included band-specific results in the main text as a complementary analysis to mirror exactly the analysis for low-frequency peaks. Accordingly, we modified the Results and Discussion sections. In the Results section; we now present the sub-section “Spatial Gradients are frequency specific” in more depth and with new plots.

Some methodological aspects have additionally been raised including addressing the question of spatial leakage, and the possible confound of inter individual differences.

On this point, we believe that we have addressed the reviewers’ comments by clarifying the points in more details and providing further evidences and analyses in the corresponding sections.

Reviewer #1:Mahjoory et al. present new results on a publicly available, resting-state MEG recording dataset from a large pool of participants. This well-written manuscript provides a systematic analysis of the spatial distribution of high-power, low-frequency, oscillations, based on non-invasive recordings of healthy, human participants. The manuscript is interesting in that the authors go beyond the mere temporal characterization of neural activity, using advanced source reconstruction approach and linear mixed effect modeling for statistics. Based on their results, the authors report a posterior-anterior spatial gradient of the dominant peak frequency, and further our understanding of the link between structure and function.Numbered summary of any substantive concerns1) My main concern regards the authors' statement that there is a decrease of the intrinsic resonance frequency from early sensory to higher-order area.i) In their analysis, the authors identify PF on the 1/f-free power spectra and then select the PF that has the highest power based on the original power spectra. Isn't this approach necessarily increasing the probability to detect a pic at low (around alpha) frequencies? In other words, such decrease in oscillations' frequency is actually a decrease of, specifically, alpha oscillations. The band-specific PF analysis actually suggests such conclusion.

Yes, we agree with the reviewer in that our results mainly shows the spatial gradients of low frequency (theta and alpha), which originates from the fact that we determine peaks from 1/f-free spectrum and define the PF as the peak with the largest amplitude in the original spectrum, which leads to the preference for low frequencies. In the revised manuscript, we present the results for both PF and band-specific PFs.

We have further analyzed (also in response to other comments) spatial gradients of the bandspecific PFs (theta, alpha, and beta) and their correlation with parameters of the 1/f component as well as cortical thickness. These analyses together with a new figure (Figure 6) are now included in the main text to show the results in more detail. Interestingly, in agreement with the main PF gradient, we found similar significant gradients (positive for theta and beta, negative for beta) of the band-specific PFs along the posterior-anterior axis. Overall these analyses support the notion that the posterior-anterior axis constitutes a principal dimension for frequency changes (both PF and band-specific PF) in the cortex.

ii) It has been shown before (e.g. Rosanova et al., 2009) that different lobes of the brain have different "intrinsic resonance frequencies," e.g. beta frequency appears in anterior regions, and the proposed analysis does not seem sensitive to it. The Band-specific PF analysis, however, shows that indeed beta increases from posterior to anterior regions.

Indeed, Rosanova (and others) have shown that different brain areas have different resonance frequencies. Similarly, in a previous study from our group we have demonstrated that different brain areas have characteristic spectral signatures (Keitel and Gross, 2016). Notably, the approach and aim of these previous studies is very different to ours. These previous studies focused on oscillatory power and treated each brain area separately, here, we focus on oscillatory frequency and study systematic changes in all spatial dimensions. So, we are indeed sensitive to different aspects of brain oscillations. This is also the case because Rosanova et al., 2009 used transcranial magnetic stimulation (TMS) to perturb directly different regions of the human cortex and then record the TMS-evoked brain response. We do not use a perturbation approach but rather study the resting-state brain activity.

We also agree with the reviewer that – irrespective of the different approaches – results of our band-specific PF analysis are consistent with findings from Rosanova and others. Also, we acknowledge that the term “intrinsic resonance frequency” can be confusing. To address the reviewer’s comment and avoid confusions for readers, we replaced “intrinsic resonance frequency” with “dominant frequency” in the revised manuscript.

iii) Based on the results reported in Figure 4A, the authors suggest that there is a change in PF and CT as a function of the region's function (i.e. sensory vs. associative regions). Yet, the data seem to suggest that the effect is driven by the location of the different regions: we see high PF in VIS and DAN, which are mainly located in the posterior part of the brain, and low PF in the other networks. And doesn't this interpretation fit better the fact that there is a link between CT and PF? In other words, change in CT are probably not sparse across the head as some of these networks seem to be, right?

We agree with the reviewer. There is a change of PF and CT associated with location and function (sensory versus associative). Both PF and CT show a posterior-anterior gradient. At the same time higher order associative brain areas are located more anterior compared to sensory brain areas (most prominently in the visual domain). Networks that are mostly localized in posterior brain areas (such as VIS) are expected to show different PF and CT compared to more anterior networks (such as DAN) merely by virtue of their location. Therefore, we agree with the reviewer that changes in PF and CT are not sparse as some of the networks are. Still, we believe that the “network representation” of our results is important to emphasize the point that these quantities (PF and CT) differ between networks. This is especially important for studies focusing a-priori on these anatomically defined networks and where these differences (in PF and CT) may confound the results.

However, we have also carried out further extensive analysis to test if PF and CT differ between networks irrespective of their location. This was done by removing from both measures changes that can be explained by linear dependencies on x,y,z. Interestingly, after regressing out the effect of spatial location, a significant difference between networks remained for PF (Figure 4—figure supplement 1A) and CT (Figure 4—figure supplement 1B). This was the case when looking at the standard resting-state networks and also when testing sensory against association areas. These results indicate that, beyond a global effect of location, networks still differ significantly in PF and CT after removing linear effects of location.

These results are now described in the main text and presented in detail in Figure 4— figure supplement 1 and 2.

2) Could the authors give an intuition of the variance in ROIs' size for a given participant and across participant, as well as how the size affects the power spectra?

In our analyses, we down-sampled the individual cortical surfaces to 8196 vertices and 16384 (triangular) faces and co-registered to a finer version of the Conte69 atlas. The atlas provides anatomical labels for each vertex but not for the faces. To obtain the surface area of a given cortical area, we first assigned a label to each face based on the label of the nearest vertex (Euclidean distance was computed between centroid of the given face and all vertices) and summed across the area values of the identically labeled faces. The obtained area values are a rough approximation but fine in this case because the spatial resolution of MEG of resting state reconstructed activity and the potential co-registration inaccuracy does not really warrant extremely fine-grained spatial details. Figure 1—figure supplement 2A shows the histogram of ROI’s size for a given participant. Figure 1—figure supplement 2B represents the mean and standard deviation of the ROI size across all participants. To assess the impact of ROI size on PF, we used again a linear mixed effect model (LMEM) where PF was defined as a function of the individual ROI area, defining the participant’s ID as a random effect to control for variability between individuals. Indeed, this model assesses the impact of ROIs’ size on PF gradient at the individual level and tests its consistency across participants. We found no significant impact of ROI size on PF (t = -1, p > 0.05). Next, we asked the question whether ROIs’ size could affect the spatial gradients of PF across the cortex irrespective of the PF itself. We used LMEM according to Equation 1 of the manuscript where we included the ROIs’ size as an additional fixed effect variable. We again found no significant impact of ROIs’ size on spatial gradients of PF along the Z and Y axes (t = -1.1, p > 0.05) (Figure 1—figure supplement 2C).

3) The authors argue that there is a posterior-anterior gradient of CT, which strongly correlates with the PF gradient. From the results presented in Figure 3, it seems that the CT gradient is more strongly explained by the Z-axis (thicker in ventral than dorsal regions).

Thanks for this constructive comment. Our focus so far has been on the posterior-anterior gradient. However, there is indeed strong evidence in our data for an inferior-superior gradient for both, CT and PF. Interestingly, following our preprint, another preprint has just confirmed these two gradients for CT (Valk et al., 2020). To address the reviewer's comment, we updated the manuscript and added z axis to the result and Discussion sections as follows:

“Notably, similar to the spatial gradients of PF, LMEM uncovered a secondary inferior-superior axis of CT. Having established the organizational axes of CT, we then tested for a significant relationship between CT and PF.”

4) Could the authors give an intuition of how stable (within participant) these PF are over time (see for instance Haegens et al, 2014)?

Thanks for raising this point. There is indeed increasing evidence that peak frequency is not constant but increases over tasks with increasing cognitive demand and decreases with increasing time on task (Benwell et al., 2019). Here, we did not expect to observe this type of change because participants did not engage in a task. But to fully address this question, we reconstructed a cortical activity time course similar to our main analysis approach. To explore consistency of our results across time, we halved source time courses of all individuals and ROIs, created two groups of data, and analyzed each group separately, as described in the Materials and methods section of the manuscript. Figure 1—figure supplement 1A illustrates the distribution of trimmed-mean-PF along the cortex for the first and second halves of the data. To statistically test the significance of the observed gradients, and their consistency across individuals, we applied LMEM on PF values of each half as a function of the ROIs’ centroid coordinates and their interactions (Figure 1—figure supplement 1B). To test whether the impact of time is significant or not, we defined the time as a categorical variable (“1^st^ half”, “2^nd^ half”) and added to our LMEM, where we defined PF as a function of centroid coordinates, their interactions and time. The statistical model showed a non-significant effect of time on our results (Figure 1—figure supplement 1C). We therefore conclude that PF and its gradient are stable over time.

5) Could the authors comment on the absence of gamma oscillations?

We initially had included gamma oscillations in our analysis. But after analyzing the data, we found gamma oscillations absent in most areas of the cortex across individuals, resulting in unreliable estimates of global gradients, which hindered us in deriving a conclusive result. This is likely at least partly due to the fact that the analysis of gamma oscillations would require a specifically optimized spectral analysis that is different from the analysis of oscillations with lower frequencies (e.g. the use of multi-taper spectral analysis might be required). Since we did not want to introduce a bias into our analysis (by having spectra with different properties), we focused on low-frequency oscillations.

However, we agree with the reviewer that changes of low frequencies across the cortex, may likely be coupled with occurrence of local gamma activities. For instance, (Voytek et al., 2010) have shown that high gamma amplitude is modulated by both the theta and alpha phase, and noted that there is a topographic separation of preferred coupling such that theta PAC is larger than alpha PAC over anterior brain regions, regardless of behavioral task, using ECOG recordings of 2 participants. We would like to keep these analyses for a future study where we account for cross-frequency couplings between low- and high-frequency oscillations and explore existence of systematic patterns across the cortex.

6) Could the authors comment on the generalizability of their results given the fact that they analyzed resting-state data?

Thank you for pointing this out. We are indeed interested in exploring implications of the observed frequency gradients in different cognitive states, disease, and aging. So far, studies have demonstrated that PF shifts as a function of cognitive load (Haegens et al., 2014), experiment duration (Benwell et al., 2019), and aging (Scally et al., 2018). But since we are the first to report a full 3-D statistical model of PF gradients it is currently unknown how they change as the brain starts to engage in a task. This will be the focus of a future study.

Reviewer #2:This study investigated a novel and interesting topic on whether there is anterior-posterior gradient in the frequency of brain oscillations using source-localized MEG data. It was found that the spatial gradient of strongest peak-frequency decreases gradually and robustly along the posterior-anterior axis following the hierarchy of early sensory and higher-order areas. The manuscript is themed around a robust spatial gradient of the frequency of brain oscillations while the main finding was the spatial gradient in the frequency of alpha oscillation but not across oscillatory frequency hierarchy. I would suggest reframing the text to be better in line the findings. Further, the main analysis are mostly sound but methodological details are missing and some parts are difficult to evaluate. Also the contribution of spatial leakage as confounding factor to the results has not been sufficiently well tested and should be better controlled for.

We appreciate that the reviewer found our study ‘novel and interesting’. We have

performed further extensive analysis to address these important points in full. Analysis and results are described below in detail.

1) The strongest peak in the original power spectrum was selected. Taken the robust alpha oscillations in rest, these peaks were in the alpha-band range (8-12 Hz) as visible in Figure 1A. The results hence describe the spatial gradient of alpha oscillations in narrow frequency range of 7 to 12 Hz. Thus, I find the title of the manuscript "The frequency gradient of human resting-state brain oscillations follows cortical hierarchies" as well as structure of text somewhat misleading and suggestive of the presence of spatial oscillatory frequency hierarchy, which seems not to be the case. Also the spatial gradients of peaks in theta and beta frequencies were estimated, but these data are now only shown in the supplementary material, not thoroughly analysed and only discussed very superficially. It looks like these data were added only for the reviewers. I think the authors should decide do they want to present a frequency gradient or an alpha-frequency gradient and modify the manuscript accordingly.

We agree with the reviewer that spatial gradients are most pronounced at low frequencies. We now state that more clearly in the revised manuscript. But, as the reviewer correctly noted, we also show significant gradients in other frequency bands. In response to this comment we have now (1) included band-specific results (including results of new analysis) as a new subsection in the main text, (2) explained the details of analysis in the Materials and methods section, and (3) discussed the results in the Discussion section. In the Results section we now present the sub-section “Spatial Gradients are frequency specific” in the main text together with a new figure (Figure 6). Moreover, we thoroughly analyzed the spatial gradients of the band-specific PFs (Figure 6) and their correlation with parameters of the 1/f signal (Figure 6—figure supplement 1) and with cortical thickness (Figure 6—figure supplement 2). Interestingly, our results show that, just as for the PF, the spatial gradients of the band-specific PFs follow the posterior-anterior axis. However, this gradient is increasing from posterior to anterior for theta and beta range oscillations but decreasing for alpha band.

2) The correlation of the frequency and cortical thickness was estimated to investigate if the frequency is correlated rather with cortical hierarchy than spatial location. A correlation of -0.14 was found (Figure 3B). Albeit very significant, the correlation itself is very weak as also shown by the large scattered peak frequencies. Compared to correlation of -0.84 of the peak frequency with posterior-anterior gradient, this result indicates that alpha peak frequency is not strongly related to cortical hierarchy. The Results and Discussion should be modified to emphasize this point. Moreover, the authors could perform a partial correlation analysis among the peak frequency, posterior-anterior direction and cortical thickness – does the correlation with thickness persist when the posterior-anterior gradient is factored out?

The low correlation actually reflects the fact that correlation analysis is not optimal in this case. It does not take into consideration the variability between participants (e.g. the fact that some participants have an overall higher occipital alpha frequency compared to others). This is also one of the reasons for the large scattering of values in Figure 3—figure supplement 2. Therefore, LMEM is, again, the preferred statistical tool because it models specifically the interindividual differences and, as a result, leads to more robust and highly significant results. Since both are a bit redundant and LMEM is more appropriate we now moved the correlation analysis to a supplement figure (Figure 3—figure supplement 2). We still included Figures 3—figure supplement 2, and the correlation results because they are intuitive and easily accessible to the readers.

We had already addressed the reviewer’s second point. Factoring out the dependency on location (x,y,z) from PF and CT still preserves the significant association between cortical thickness and peak frequency (Figures 3—figure supplement 1). This is indeed an important result because it suggests that PF and CT are related in a way that cannot be explained by a common linear dependence on location.

3) Related to the above, in Materials and methods the authors write that to obtain correlation between PF and CT scores, the robust correlation was performed (Pernet et al., 2012). Pernet et al. released a new open source Matlab toolbox for analysis of robust correlations using multiple statistical tests. Which of tests would the authors have used for the correlation analysis?

After plotting the data (scatter plot) and inspecting all correlation alternatives offered by Robust Correlation Toolbox, we chose Skipped Pearson Correlation for our analysis because it captured most of the variation in the data. However, all alternative techniques also led to similar results. To address this comment, we updated the manuscript and added further details for the applied correlation analysis.

Materials and methods section:

“To obtain the correlation between PF and CT scores, we initially computed the 10% trimmed mean across participants for each ROI and performed a robust correlation (skipped Pearson correlation, Robust Correlation Toolbox) (Pernet et al., 2012)”

4) The observed spatial gradient of peak frequencies can reflect a true continuous frequency variation along an anatomical axis or, alternatively, it could be caused by source leakage of distributed alpha sources at different frequencies and different cortical areas. The authors argue that if the spatial gradient is due to leakage then there should not be frequency change in alpha oscillations in V1. As there was a significant correlation of PF of V1 and its nearby 0.5-1.5 cm sources, the authors concluded that there is no source-leakage from the frontal source that could explain the spatial gradient. However, the alpha sources that could become mixed and cause confounding the results do not need to be localized to V1 and frontal cortex and hence estimating the source-leakage of V1 alpha source appears inadequate in my opinion. Classical MEG alpha literature suggests that the strongest alpha sources could actually be located higher in the visual cortical hierarchy, in the parieto-occipital sulcus or in areas near precuneus which could well cause significant leakage affecting the analyses. Second, the spatial accuracy of source-localized MEG is > 1-2 cm and hence the area for which the source-leakage was estimated could have been too small to test whether there is leakage or not.

We agree with the reviewer in that alpha generators could be distributed at different frequencies and different cortical areas. However, we believe that our analysis is relatively insensitive to the exact reference location for the following reason: The simplest scenario for a gradient caused by leakage consists of two sources of different frequencies (for example an occipito-parietal alpha source and a frontal theta source). Differential leakage of occipital and frontal sources at different points on the posterior-anterior line could result in a gradient. To address this point, according to the reviewer’s valid comment, we computed the geodesic distance between the reference ROI, V1, and all areas located 2–3cm away from V1, and applied linear mixed effect modelling of PF as a function of the distance values. We found a highly significant negative dependence between PF and distance (t = -21.11, p << 0.001).

To further account for the potential confounding effect of spatial leakage we performed a new comprehensive analysis, where we computed the geodesic distance between centroid of all ROIs and the centroid of V1 and used this as a new axis because it well explains the posterior-anterior axis in the cortex (Figure 1D bottom panel). We applied an LMEM between the PF as a response variable and the geodesic distance values as an independent variable to assess the distribution of PF along the distance values while considering inter-individual variability. We found a highly significant gradient of PF along the specified geodesic distance (t = -18.9, p << 0.001). Furthermore, to answer the question whether the spatial gradient of PF constantly exists in different areas of the cortex, we split the cortex to 3 equal, consecutive and non-overlapping windows (about 4 cm) based on the Y axis, applied LMEM for each window modelling PF as a function of geodesic distance, and found a significant gradient (window 1: t = -8.1, window 2: t = -10.9, window 3: t = -5.2, all p < 0.001) (Figure 1E). Indeed, our analysis demonstrates a significant organization of PF along the posterior-anterior direction for all windows indicating that this axis constitutes a systematic and constant gradient of PF. We updated the manuscript as follows:

“To address this question, we computed the geodesic distance between the reference ROI, V1, and all areas located 2–3 cm away from V1, and applied linear mixed effect modelling of PF as a function of the distance values. […] Indeed, our analysis demonstrates a significant organization of PF along the posterior-anterior direction for all windows indicating that this axis constitutes a systematic and constant gradient of PF.”

5) The authors have used a Beamforming approach for source reconstruction. Several details of the source reconstruction approach are missing or unclear. First, if Beamforming was used for the entire brain volume in voxel space, how was data transformed to cortical parcellations and how were the values for each parcel obtained? The sensor covariance matrix was computed for 2 sec trials. Which trials were these?

In the revised manuscript we have added further details. Beamforming was used not in the entire brain volume but directly on vertices of the cortical surface. For each anatomical ROI we performed dimensionality reduction using singular value decomposition (svd) on all vertex timeseries. We retained the required number of components to account for 95% of the variance for each ROI (typically 15 components). The final spectrum was computed as the 10% trimmed mean across spectra. The 2-sec trials refer to data segments obtained from cutting the continuous data into 2-sec long segments. These details have now been further clarified in the revised manuscript. The relevant parts read:

“[…] Using the Caret software, the mid-thickness cortical surface (halfway between the pial and white matter surfaces) was generated. The cortex surface was parceled into 384 ROIs (192 per hemisphere) according to Schoffelen et al. (Schoffelen et al., 2017). 648 vertices located in the medial wall (sub-cortical areas) were excluded from further analysis. […] The sensor covariance matrix was computed between all MEG-sensor pairs, as the average covariance matrix after cutting the continuous data into 2-second data segments. […] For each anatomical ROI we performed dimensionality reduction using singular value decomposition (SVD) on all vertex timeseries. We retained the required number of components to account for 95% of the variance for each ROI (typically 15 components). Component time courses were segmented to twosecond epochs, from which power spectra were computed using a multi-tapered Fast Fourier transform, using discrete prolate spheroidal sequences (dpss) as windowing function, with 2 Hz spectral smoothing. To obtain a single spectrum for each ROI (ROI spectrum), we pooled spectra of epochs across components and computed the 10% trimmed mean across them. Averaging after leaving out 10% of data from left and right tails of the spectra distribution offers a more robust estimate.”

Reviewer #3:Mahjoory et al. examined a large dataset of human MEG-recorded neuronal oscillations to probe the relation between oscillation properties across space and cortical thickness. Their main findings are that the brain shows anterior-posterior gradients in the frequency of neuronal oscillations and in the slope of 1/f nonoscillatory MEG patterns. They also find a matching A-P gradient in cortical thickness, and they suggest that cortical thickness correlates closely with oscillation frequency in individual subjects, even after accounting for mean anatomical variations in each pattern.My enthusiasm for publishing this paper at eLife is limited because most of their findings are not especially novel. Specifically, it is already known that oscillation frequency varies across the brain (see work by Voytek et al., Groppe et al., Zhang et al., and others) and merely replicating this finding in a large open dataset is not extremely innovative. Similarly, Figure 5 on oscillatory peaks is not novel and neither is the result showing an overall A-P gradient in mean cortical thickness.

The reviewers state that it is already known that the frequency of oscillations changes across the brain. This is correct and we acknowledge that in our manuscript. However, this is not the main result of our study. The novel contributions of our study are the following:

(1) We provide the first comprehensive (in space and frequency) statistical model of frequency gradients in a large resting-state data set. The main result is not that frequency changes across the cortex but that it changes systematically and globally along spatial (and hierarchical) gradients. To the best of our knowledge, the only study that had previously tested this hypothesis is Zhang et al., 2018. However, their result was based on a smaller sample, data was recorded from epilepsy patients with inherent pathological brain activity and the ECoG data was constrained by limited electrode coverage. In addition, only anterior-posterior changes were studied. We provide the first full 3D statistical model at the level of individual brain areas (showing for example a significant gradient in inferior to superior direction).

(2) We provide the first full 3D statistical model of cortical thickness in a large data set and show that cortical thickness changes systematically in space and is correlated with peak frequency.

(3) We show for the first time that frequency gradients follow cortical thickness (as a proxy of hierarchical level) more closely than can be explained purely by spatial location.

We believe that the relevance of these three novel findings based on state-of-the-art statistical models in a large data set meet the criteria for this prestigious journal.

The paper's novel claim is showing that oscillation frequency correlates with cortical thickness even after accounting for mean anatomical patterns. However, this result is not adequate to justify publishing the entire paper, especially because the data underlying this result were not examined in much detail and there was no compelling and specific proposed mechanism to directly link these two phenomena. Further, I am concerned that this apparent correlation could actually be a reflection of intersubject differences in mean thickness and oscillation properties, rather than by a detailed region-by-region correspondence between these variables within-subject, as the authors suggest. Failing to rule out this possibility is a substantial weakness in this analysis and the authors could do more to demonstrate this effect at the within-subject level.

The correlation of oscillations frequency and cortical thickness is not the only novelty but only one of three main novel results (as listed in our response to the previous comment). In addition, we can rule out the main concern of the reviewer that the correlation reflects intersubject variability. This is an important concern but it is already addressed in our analysis. First, we made sure that our analysis does not introduce unnecessary variability across participants. Our source analysis and subsequent parcellation was based on a standard anatomical atlas and the procedure ensured that the number of vertices for a given labelled area is the same across participants. Second, and more importantly, our use of linear-mixed-effect models (LMEM) ensured that individual differences are properly accounted for and that significant gradients are consistently present in individual participants. For example, participants have different alpha peak frequency in occipital brain areas and the slope of the frequency gradient is different. These individual differences are specifically modelled by LMEM as random effects. Importantly, LMEM applies two-level statistics, and therefore, will only show a significant gradient if it is significant across cortical areas at the individual level, and consistent across participants at the group level. We have further corroborated the existence of spatial gradients by computing linear correlations for each individual and then analyzing individual correlations across the group. At the individual level, we computed the Spearman correlation between Y coordinates and PF values. For group level analysis we first computed the inverse hyperbolic tangent of the obtained correlation values, and applied one sample t-test across them. This approach tests the hypothesis that, across the group, correlations deviate significantly from zero. The results showed a highly significant negative correlation (t-value = -15.52, p < 0.001) between PF and Y coordinates. As expected, the results are consistent with the LMEM results. Both approaches show significant gradients at the individual level. However, we use LMEM throughout the manuscript because it is the statistically superior and more versatile approach.

[Editors’ note: what follows is the authors’ response to the second round of review.]

Revisions:1) A key part of the paper is showing frequency gradients both within and across subjects. The rebuttal letter does a good job explaining that the authors' statistical framework identifies this pattern robustly both within and across subjects. But the text related to this is still unclear. The authors should revise the text of the results to more clearly explain how their statistical framework identifies gradients both within and across subjects.

To address the mentioned points we updated the following sections of the text:

We updated the first subsection of the result section, adding more details about the significance of the posterior-to-anterior gradient of PF at individual level across ROIs and its consistency across individuals.In the result section, we now describe further details about the LMEM and how it deals with the intra- and inter-individual variability.We added new plots to Figure 1, which illustrate the correlation between PF and Y-coordinates at the individual level and their distribution across subjects.We updated the Discussion section further discussing the inter-individual variability.

2) The authors should also consider revising their figures to show clear examples of within and across subject gradients. The scatter plots and brain plots in Figure 1, for example, are hard to understand because they seem to combine data both across subjects and regions. It would be very informative if the figures followed the statistical results.

Thanks for suggesting this important point, which indeed requires further clarification. To illustrate within- and across-individual gradient we added new plots to Figures 1B and 1D, top-right panel, to display the individual correlation values across ROIs and to show the distribution of correlation values across all participants.

3) In response to one of the reviewers, the revised paper now includes an analysis of analyses within specific bands. This new analysis is a bit hard to follow in the context of the paper because it is unclear how it relates to the paper's primary analyses. Is the idea that there are multiple oscillatory patterns at different frequencies that all show gradients simultaneously? Additional clarity here would be helpful.

In our analysis pipeline, we obtained power spectrum for each region of interest (ROI), from which we determined all spectral peaks at the range of 3–45 Hz. This analysis led to identification of multiple peaks for (some) ROIs. Figure 5 represents a histogram plot of all identified peaks pooled across all ROIs and individuals. The plot nicely delineates canonical oscillatory frequency bands. We then extracted peaks, separately, within each band for each participant and ROI and repeated the LMEM analysis. Indeed, this revealed existence of band-specific gradients in addition to the dominant peak frequency gradient. This means that peak frequencies change consistently across space in these canonical frequency bands. We have now clarified this point in the Results and Discussion section. In addition, we now describe more clearly the term “peak frequency” which also helps to clarify the distinction between gradient of peak frequency and gradients of band-specific peaks.

**References**

Benwell, C.S.Y., London, R.E., Tagliabue, C.F., Veniero, D., Gross, J., Keitel, C., Thut, G., 2019. Frequency and power of human alpha oscillations drift systematically with time-on-task. Neuroimage 192, 101–114. doi:10.1016/j.neuroimage.2019.02.067

Haegens, S., Cousijn, H., Wallis, G., Harrison, P.J., Nobre, A.C., 2014. Inter- and intra-individual variability in alpha peak frequency. Neuroimage 92, 46–55. doi:10.1016/j.neuroimage.2014.01.049

Scally, B., Burke, M.R., Bunce, D., Delvenne, J.-F., 2018. Resting-state EEG power and connectivity are associated with alpha peak frequency slowing in healthy aging. Neurobiol. Aging 71, 149–155. doi:10.1016/j.neurobiolaging.2018.07.004